# Training More Robust Classification Model via Discriminative Loss and Gaussian Noise Injection

**Hai-Vy Nguyen**                                                                    *hai-vy.nguyen@ampere.cars*
*Ampere Software Technology – Institut de mathématiques de Toulouse*
*– Institut de Recherche en Informatique de Toulouse*

**Fabrice Gamboa**                                                    *fabrice.gamboa@math.univ-toulouse.fr*
*Institut de mathématiques de Toulouse*

**Sixin Zhang**                                                                          *sixin.zhang@irit.fr*
*Institut de Recherche en Informatique de Toulouse*

**Reda Chhaibi**                                                              *reda.chhaibi@univ-cotedazur.fr*
*Laboratoire Jean Alexandre Dieudonné, Université Côte d'Azur*

**Serge Gratton**                                                              *serge.gratton@toulouse-inp.fr*
*Institut de Recherche en Informatique de Toulouse*

**Thierry Giaccone**                                                      *thierry.giaccone@ampere.cars*
*Ampere Software Technology*

**Reviewed on OpenReview:** *https://openreview.net/forum?id=RnLfJgvST2*

## Abstract

Robustness of deep neural networks to input noise remains a critical challenge, as naive noise injection often degrades accuracy on clean (uncorrupted) data. We propose a novel training framework that addresses this trade-off through two complementary objectives. First, we introduce a loss function applied at the penultimate layer that explicitly enforces intra-class compactness and increases the margin to analytically defined decision boundaries. This enhances feature discriminativeness and class separability for clean data. Second, we propose a class-wise feature alignment mechanism that brings noisy data clusters closer to their clean counterparts. Furthermore, we provide a theoretical analysis demonstrating that improving feature stability under additive Gaussian noise implicitly reduces the curvature of the softmax loss landscape in input space, as measured by Hessian eigenvalues. This thus naturally enhances robustness without explicit curvature penalties. Conversely, we also theoretically show that lower curvatures lead to more robust models. We validate the effectiveness of our method on standard benchmarks and our custom dataset. Our approach significantly reinforces model robustness to various perturbations while maintaining high accuracy on clean data, advancing the understanding and practice of noise-robust deep learning.

## 1 Introduction

Deep neural networks have achieved remarkable success across a wide range of tasks. However, their instability to input perturbations, including noise and adversarial attacks, remains a major issue (Szegedy et al., 2014; Goodfellow et al., 2015). Training models to be robust against such perturbations is crucial for deploying reliable machine learning systems in real-world noisy environments. For instance, when a classification model is trained only with uncorrupted data, it tends to produce features that are not well separated between classes when facing noisy data (Fig. 1a).

A common and straightforward approach to improve robustness is to augment the training data by adding some noise (Bishop, 1995). While noise injection often improves model robustness, it can also degrade the accuracy on the uncorrupted (clean) data. This trade-off arises because naively forcing the model to fit

noisy data generally reduces the discriminativeness of the learned features. This results in blurred decision boundaries and ambiguous feature representations (Fig. 1b).

Therefore, naively injecting noise into the training data is inadequate and may adversely affect the model's performance on the original data distribution. This observation motivates the need for principled methods that explicitly balance robustness to noisy inputs with the preservation of discriminative feature representations for original data (features produced by our method are depicted in Fig. 1c).

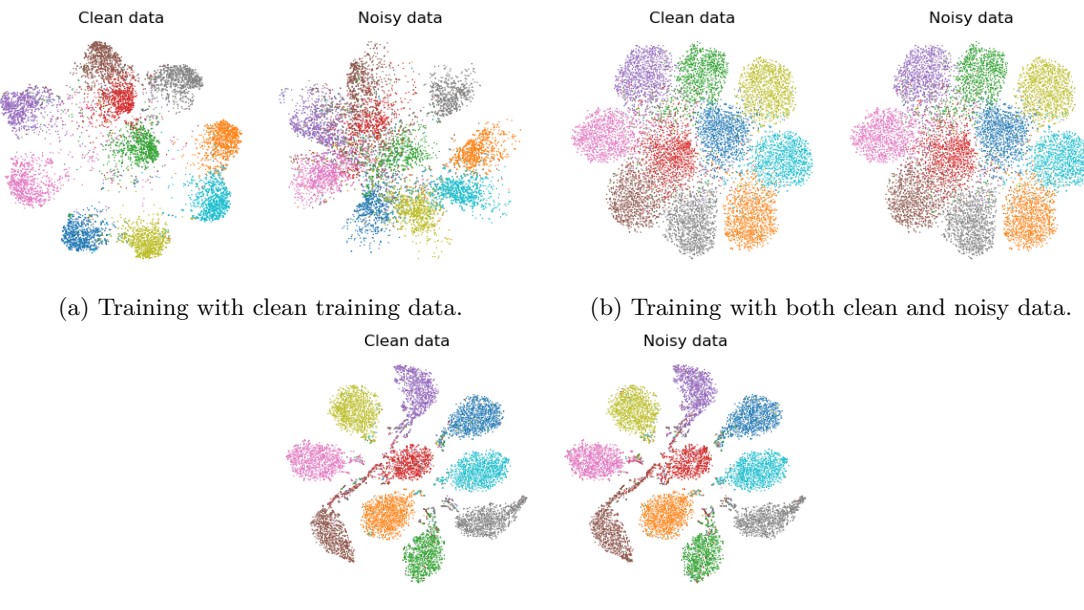

Figure 1: t-SNE feature visualization for test set of CIFAR10 of original (clean) and noisy data (additive Gaussian noise), produced by models trained with different methods. When training the model with the clean data, it produces features that are not well separated between the classes for noisy data (right plot of Fig. 1a). Training model with both clean and noisy data helps to produce more discriminative features for the noisy data (Fig. 1b) but also makes features on the clean data less discriminative. Our method helps the model to produce more discriminative features both on clean and noisy data (Fig. 1c).

In this paper, we propose a novel framework that explicitly addresses these challenges by focusing on two complementary objectives (illustrated in Fig. 2):

1. **Boosting the discriminativeness of features for clean data.** We introduce a new loss operating at the penultimate layer of the network, which encourages intra-class compactness and simultaneously increases the margin between features and decision boundaries. Crucially, in this penultimate feature space, decision boundaries can be analytically characterized as hyperplanes. This enables an explicit and tractable formulation for the loss. Consequently, by directly shaping the geometry of the feature space, our loss strengthens class separability, mitigating the blurring effect that noise injection may induce.

2. **Aligning noisy data features with clean data clusters.** Unlike classical approaches that enforce individual noisy samples to be close to their clean counterparts (e.g., stability training (Zheng et al., 2016) or Lipschitz-constrained networks (Tsuzuku et al., 2018)), our method performs *class-wise* alignment. Specifically, we encourage the entire cluster of noisy samples for a given class to be close to the corresponding cluster of clean samples from the same class. This cluster-level alignment is less restrictive and allows the model to maintain expressiveness, avoiding the performance degradation often observed when forcing strict pointwise matching between noisy and clean samples.

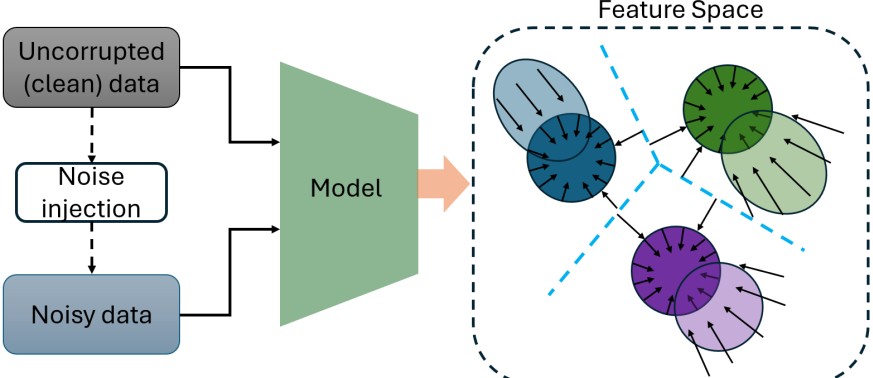

Figure 2: Illustration of our method applied on the features of the penultimate layer. In the feature space, each color represents the data cluster of each class, where darker and lighter colors represent clean and noisy data, respectively. Our method focuses on boosting discriminativeness of features on clean data (by enforcing intra-class compactness and inter-class separability) and aligning noisy data clusters with those of clean data from the same class (in feature space).

**Theoretical insights on noise injection and curvature.** Beyond the novel loss function design, we provide a theoretical analysis revealing that training with Gaussian noise implicitly acts as a loss curvature regularizer in the input space (that we call *input loss curvature*). More precisely, we show that noise injection reduces the eigenvalues of the Hessian of the loss with respect to the input, effectively smoothing the local loss landscape. This curvature reduction serves as a natural regularization mechanism that enhances robustness against input perturbations without explicitly adding curvature penalty terms. Conversely, we also show that a lower curvature in the input space leads to a model producing more stable features under Gaussian perturbations. This insight bridges the gap between noise-based robustness methods, which traditionally rely on data augmentation or smoothing, and curvature-based approaches that explicitly constrain the geometry of the loss surface (Moosavi-Dezfooli et al., 2019). Our work is, to the best of our knowledge, the first to formally link Gaussian noise training, curvature reduction in input space, and improved robustness in a unified framework.

**Contributions.** The main contributions of this paper are summarized as follows:

- We propose a novel loss function applied at the penultimate layer that simultaneously enforces intra-class compactness and maximizes inter-class margins with respect to analytically defined decision boundaries (hyperplanes). This encourages more discriminative feature representations for clean data. We also provide theoretical insights showing that enforcing only intra-class compactness or only inter-class separability can lead to trivial solutions that fail to improve model performance (Proposition 5.3). This justifies our design choice of combining both constraints.

- In addition to the loss applied on clean data, we introduce a simple yet effective class-wise feature alignment strategy that encourages clean and noisy data clusters to align in the feature space. This contrasts with classical point-wise stability methods and helps preserve model expressiveness, avoiding the performance degradation commonly seen with naive noise augmentation (see our method in Fig. 1c).

- We theoretically demonstrate that training with additive Gaussian noise reduces the eigenvalues of the input-space Hessian of the standard cross-entropy loss, effectively regularizing curvature. Conversely, we also prove that reduced curvature in the input space leads to more stable features under Gaussian noise, from a probabilistic standpoint (Theorem 6.1).

- We provide further theoretical insights into the proposed method. This includes a generalization behavior analysis for the compactness constraint (Theorem 6.2), and an investigation of how applying the loss solely at the penultimate layer impacts intermediate representations (Theorem 6.3).

- We demonstrate the theoretical insights and validate the effectiveness of our method through experiments on both standard benchmarks and our road image dataset (Section 7). Notably, models trained exclusively on Gaussian-noised data using our approach show substantial performance gains over baseline methods when evaluated on other perturbations, such as random occlusion and resolution degradation (see Fig. 7 for examples).

**Paper Organization.** Section 2 reviews related works. Section 3 introduces the standard softmax classification framework and the notations used throughout the paper. It also discusses theoretical decision boundaries and the limitations of the standard softmax loss. Building on this, Section 4 presents our proposed method, followed by a discussion of its design properties in Section 5. Theoretical insights into our method are provided in Section 6, including the impact of our method on input-space curvature under Gaussian noise (Section 6.1), generalization analysis of the compactness constraint (Section 6.2), and the effect of our method on intermediate representations (Section 6.3). Experimental results on standard and custom datasets are presented in Section 7. Our code is publicly available at code-robust-training.

## 2 Related works

**Adversarial perturbations and Lipschitz models.** Training models that are more stable against input perturbations is an important subject in deep learning. A well-studied direction is to include examples with adversarial perturbations in training (Dong et al., 2020; Miyato et al., 2016; Qin et al., 2019). While this approach has proven its effectiveness, it is very time-consuming to generate adversarial perturbations. Another direction is to construct models with Lipschitz property so that features of the corrupted input stay close to those of clean counterpart (Fazlyab et al., 2023; Zhang et al., 2022; 2021). This approach requires to modify the model architecture (to ensure Lipschitz property by construction), so it is not applicable on an existing base model.

**Noise Injection.** Training neural networks with noise injected into inputs or weights is a classical technique for improving generalization (e.g., Bishop (1995); Zheng et al. (2016); He et al. (2019)). Bishop established that input noise training is approximately equivalent to Tikhonov regularization in the parameter space, effectively smoothing the learned function. However, this equivalence does not explicitly connect noise injection to changes in the curvature of the loss landscape in the input space. More recent research has leveraged noise during inference to improve robustness against perturbations such as the method of randomized smoothing (Cohen et al., 2019; Levine & Feizi, 2020; Scholten et al., 2023). This method is more computationally expensive, as it requires multiple inference passes per data point to average the model outputs over Gaussian noise added to the inputs. Moreover, it does not directly analyze or control the curvature of the loss function with respect to the inputs.

**Loss curvature in Deep Learning.** Understanding the geometry and curvature of the loss landscape in deep learning has been an important area of research for understanding the model stability. These studies mainly focus on the model parameter space and reveal that sharp or flat minima in the parameter space can significantly affect a model's generalization performance (see, e.g., Foret et al. (2021); Dinh et al. (2017); Li et al. (2018); Andriushchenko & Flammarion (2022)). However, the relationship between the model stability and the loss curvature in the input space seems to be less studied. Moosavi-Dezfooli et al. Moosavi-Dezfooli et al. (2019) demonstrate that the Hessian of the loss function with respect to the input captures local curvature information that is essential for understanding a model's sensitivity to perturbations. Several works have explored explicit regularization strategies targeting curvature metrics, such as Hessian norms of the model output or loss with respect to the inputs (Mustafa et al., 2020; Moosavi-Dezfooli et al., 2019). Most of these approaches requires the explicit computation or approximation of the Hessian, which can be computationally expensive. Other related methods focus on Jacobian regularization (Rifai et al., 2011) or smoothing techniques for boosting local linearity (Qin et al., 2019) that implicitly reduce sensitivity to input perturbations but do not necessarily analyze the eigenvalues directly.

**Discriminativeness in feature spaces.** To enhance the discriminativeness of learned representations, Tang (2013) proposed a margin-based approach that integrates multi-class support vector machines (SVMs) into deep networks. Classical SVMs and the deep SVM formulation of Tang (2013) essentially maximize

inter-class margins via objectives defined over separate linear classifiers. This is done by replacing the softmax layer with $C$ one-vs-rest SVMs, introducing additional classifier parameters and slack variables. This setup enforces the margin at the classifier output level but does not directly constrain the geometry of the features. In contrast, our approach retains the standard softmax layer and instead defines, analytically, the decision boundaries induced by this softmax classifier in the feature space. This enables us to impose margin-based constraints and intra-class compactness directly on the learned features without introducing additional classifiers or slack parameters. Elsayed et al. (2018) similarly advocate maximizing the margin to decision boundaries in feature space via a quadratic approximation. However, their formulation focuses solely on the approximated Euclidean margin and does not impose constraints on intra-class compactness. In contrast, our loss explicitly incorporates intra-class compactness, which plays a crucial role in encouraging smaller curvature, as analyzed in our theoretical section. In another line of work, Wen et al. (2016) introduced the *center loss*, which explicitly encourages intra-class compactness by minimizing the distance between each feature and its corresponding class centroid. However, this formulation does not explicitly enforce inter-class separability. The advantages of large-margin learning in deep neural networks were further emphasized by Liu et al. (2016) and Liu et al. (2017). More recently, Papyan et al. (2020) observed that within-class variation tends to vanish if one continues training a softmax-based classifier beyond the zero-error regime. While this finding is theoretically appealing, achieving the zero-error regime in practice is challenging. Therefore, it is often desirable to introduce supplementary discriminative constraints, as in the aforementioned works. In our method, we propose to simultaneously enforce intra-class compactness and inter-class separability to enhance feature discriminativeness when training with noise-injected inputs.

**Our contribution compared to related literature.** In contrast to prior work, our method theoretically and empirically demonstrates that training with Gaussian noise inherently leads to a reduction in the eigenvalues of the Hessian of the loss function with respect to the inputs. This mechanism acts as an implicit curvature regularizer, naturally flattening the local loss landscape without the need for explicit penalty terms. Conversely, we also demonstrate that the curvature reduction correlates with robustness to Gaussian noise perturbations at inference time. This provides a principled explanation for why noise injection enhances robustness beyond classical generalization arguments. To the best of our knowledge, this explicit link between noise injection during training, input loss curvature reduction, and improved noise robustness has not been previously characterized in the literature. Moreover, our method also emphasizes enhancing the discriminative power of clean data features to ensure strong performance on clean inputs. This is in contrast with standard approaches, where one typically focuses only on improving the accuracy on noisy (corrupted) data. Our paper thus offers both practical and theoretical insights that complement and extend existing approaches.

## 3 Background and framework

### 3.1 Preliminaries: classification with softmax model

Let us consider a classification problem with $C$ classes ($C \geq 2$). The input space is denoted by $\mathcal{X}$, which can be a space of images, time series or vectors. The neural network (backbone) transforms an input into a fixed-dimension vector. Formally we model the network by a function: $f_\theta : \mathcal{X} \mapsto \mathcal{F} \subseteq \mathbb{R}^d$, where $\theta$ is the set of parameters of the neural network, $d$ is the dimension of the so-called *feature space* $\mathcal{F}$. That is, $\mathcal{F}$ is the representation space induced from $\mathcal{X}$ by the transformation $f_\theta$. Given an input $x \in \mathcal{X}$, let $q = f_\theta(x)$. In order to perform a classification task, $q$ is then passed through a softmax layer consisting of an affine (linear) transformation (Eq. (1)) and a *softmax* function (Eq. (2)). Expressed in formal equations, we have

$$z = Wq + b, \ W \in \mathbb{R}^{C \times d} \text{ and } b \in \mathbb{R}^C \ , \tag{1}$$

$$\sigma(z)_i = \frac{e^{z_i}}{\sum_{j=1}^{C} e^{z_j}} \ . \tag{2}$$

Here, for $i = 1, \cdots, C$, $z_i$ (resp. $\sigma(z)_i$) is the $i^{th}$ component of column-vector $z$ (resp. $\sigma(z)$). The components $z_i$'s of $z$ are called logits (and so $z$ is called logit vector). The predicted class is then the class with maximum

value for $\sigma(z)$, i.e.

$$\widehat{y}(x) = \arg\max_i \sigma(z)_i.$$

In summary, the whole softmax model can be summarized as the following sequence of transformations in Fig. 3.

$$x \in \mathcal{X} \xrightarrow{\quad f_\theta \quad} q(x) \in \mathbb{R}^d \xrightarrow{\text{Affine } (W,b)} z(q) \in \mathbb{R}^C \xrightarrow{\text{softmax function}} \sigma(z) \in \Delta^{C-1} \subset \mathbb{R}^C$$
$$\underbrace{\qquad\qquad\qquad\qquad\qquad\qquad\qquad\qquad\qquad\qquad}_{\mathcal{N}_\Theta}$$

Figure 3: Pipeline from input $x$ to softmax output $\sigma(z) \in \Delta^{C-1}$, which is the simplex in $\mathbb{R}^C$ of probability measures. The whole softmax model is denoted by $\mathcal{N}_\Theta$, composed of $f_\theta$ and the softmax layer, where $\Theta$ is the concatenation of all the parameters of $(\theta, W, b)$.

**Remark 3.1** *We note that* $\arg\max_i \sigma(z)_i = \arg\max_i z_i$. *So that,* $\hat{y} = \arg\max_i z_i$.

In the standard approach, to train the neural network to predict maximal score for the right class, the softmax loss is used. This loss is written as

$$\mathcal{L}_\mathcal{S}(\Theta; \mathcal{B}) = \frac{1}{|\mathcal{B}|} \sum_{(x,y)\in\mathcal{B}} l(x,y) \ , \tag{3}$$

where

$$l(x,y) = -\log \sigma \left( W f_\theta(x) + b \right)_y \ . \tag{4}$$

Here, $\Theta$ is the concatenation of all the parameters of $(\theta, W, b)$, $\mathcal{B}$ is the current mini-batch, $x$ being a training example associated with its ground-truth label $y \in \{1, 2..., C\}$. By minimizing this loss function w.r.t. $\Theta$, the model learns to assign maximal score to the right class.

Throughout this paper, $q$ and $q(x)$ refer generally to the same object. Writing $q(x)$ highlights its dependence on $x$. By analogy, depending on the context, we may write $z(q)$ in place of $z$. We also denote the whole softmax model by $\mathcal{N}_\Theta$, composed of $f_\theta$ and the softmax layer. This notation will be used in Section 5.2 for our theoretical insights.

### 3.2 Decision boundaries and the drawback of softmax loss

As presented in the last section, a feature vector $q \in \mathcal{F}$ induces a logit vector $z(q) = Wq + b \in \mathbb{R}^C$. Consider the pair of classes $\{i, j\}$, we have:

$$z_i(q) - z_j(q) = (Wq + b)_i - (Wq + b)_j = \langle W_i - W_j, q \rangle + (b_i - b_j). \tag{5}$$

Here, $z(q) = \begin{pmatrix} z_1(q) \\ \vdots \\ z_C(q) \end{pmatrix}$ and $W = \begin{pmatrix} W_1^T \\ \vdots \\ W_C^T \end{pmatrix}$. Note that for $j = 1, \cdots, C$, $W_j \in \mathbb{R}^d$. Set

$$\mathcal{P}_{ij} = \{q \in \mathcal{F} \ , \ \langle W_i - W_j, q \rangle + (b_i - b_j) = 0\} \ . \tag{6}$$

Notice that $\{q \in \mathcal{F} : z_i(q) > z_j(q)\}$ and $\{q \in \mathcal{F} : z_i(q) < z_j(q)\}$ are the two half-spaces separated by $\mathcal{P}_{ij}$. Hence, the decision boundary for the pair $\{i, j\}$ is the hyperplane $\mathcal{P}_{ij}$, and for $q \in \mathcal{P}_{ij}$, the scores assigned to the classes $i$ and $j$ are the same ($z_i(q) = z_j(q)$). Using Eq. (3), for an input of class $i$, we see that the softmax loss pushes $z_i$ to be larger than all other $z_j$'s ($j \neq i$), i.e., $\langle W_i - W_j, q \rangle + (b_i - b_j) > 0$. Hence, the softmax loss enforces the features to be in the right side w.r.t. decision hyper-planes. Notice that the softmax loss has a contraction effect. In fact, inputs from the same class tend to produce probability vectors that are close to each other and lie near an extremal point of $\Delta^{C-1}$, the simplex of probability measures

in $\mathbb{R}^{C1}$. However, this does not guarantee that features of the same class form a compact cluster in feature space. To make this precise, we introduce the following definition.

**Definition 3.1 (Class dispersion)** *We define the dispersion of a given class c as the maximal Euclidean distance (in feature space $\mathcal{F}$) between two samples belonging to that class:*

$$\text{dispersion}(c) := \max_{x_1,x_2 \in \mathcal{C}_c^{\mathcal{D}}} \|q(x_1) - q(x_2)\|,$$

*where $\mathcal{C}_c^{\mathcal{D}} = \{x : (x,y) \in \mathcal{D}, \ y = c\}$ and $\mathcal{D}$ denotes the training dataset.*

A key reason is the translation invariance of the softmax function. Specifically, for any $\varepsilon \in \mathbb{R}$, we have $\sigma(z) = \sigma(z + \varepsilon\mathbf{1})$, where $\mathbf{1}$ is the all-ones vector. Thus, for a reference feature $q \in \mathcal{F}$ with logits $z = Wq + b$, the set of features that yield the same softmax output is

$$\mathcal{S}(q) = \{ q + v : Wv = \varepsilon\mathbf{1}, \ \varepsilon \in \mathbb{R} \}.$$

This implies that even if $\sigma(z(q_1)) = \sigma(z(q_2))$, the corresponding features $q_1$ and $q_2$ can be arbitrarily far apart (unless the classifier matrix $W$ has very specific structure). Hence, although two inputs $x_1$ and $x_2$ from the same class may produce probability vectors close to each other (and close to an extremal point of $\Delta^{C-1}$), there is no guarantee that $q(x_1)$ and $q(x_2)$ are close in feature space. This is because entire subspaces of features can map to the same probability vector (illustrated in Fig. 4). This suggests the need of an explicit constraint that enforces such compactness (i.e., small *class dispersion* for all the classes).

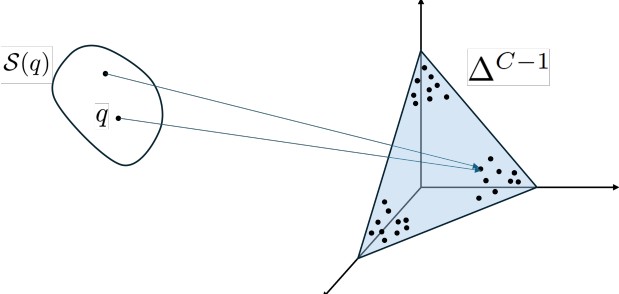

Figure 4: Motivating remark. Ideally, inputs from the same class produce probability vectors close to each other and near a corner of $\Delta^{C-1}$. However, for a given feature $q \in \mathcal{F}$, there exists an entire subspace mapping to the same probability vector. Consequently, using the softmax loss alone does *not* enforce small *class dispersion*.

# 4 Proposed method

As presented in the introduction, our method includes two complementary objectives: boosting the discriminativeness of features for clean data (Eq. (7)) and aligning noisy data features with clean data clusters (Eq. (12)). This will be introduced in Sections 4.1 and 4.2 subsequently. Note that in our method, these proposed constraints are applied alongside the standard softmax loss on clean data, as this helps produce more separable features (as discussed in Section 3). These objectives are optimized with two simultaneous forward passes corresponding to clean and noisy inputs. Although the proposed constraints are general and applicable to arbitrary noise distributions, in this work we restrict our focus to data perturbed with Gaussian noise during training.

---

[1]Formally, $\Delta^{C-1} = \{(p_1, \ldots, p_C) \in \mathbb{R}^C \mid \sum_{i=1}^C p_i = 1, \ p_i \geq 0 \ \forall i\}$.

### 4.1 Proposed loss function on clean data

To have a better classification, we rely on the following two factors: intra-class compactness and inter-class separability. To obtain these properties, we work in the feature space $\mathcal{F}$. In many classification problems, the intra-class variance is very large. So, by forcing the model to map various samples of the same class in a compact representation, the model learns the representative features of each class and ignores irrelevant details. Moreover, it may happen that samples in different classes are very similar. This leads to misclassification. Hence, we also aim to learn a representation having large margins to the decision boundaries between classes. To achieve all these objectives, we propose the following loss function on uncorrupted (clean) data:

$$\mathcal{L}_{\text{clean}} = \alpha \cdot \mathcal{L}_{\text{compact}} + \beta \cdot \mathcal{L}_{\text{margin}} + \gamma_{\text{reg}} \cdot \mathcal{L}_{\text{reg}} \ . \tag{7}$$

Here, $\mathcal{L}_{\text{compact}}$, $\mathcal{L}_{\text{margin}}$ and $\mathcal{L}_{\text{reg}}$ enforce the constraints for class compactness, inter-class separability and regularization, respectively. We now discuss in detail these three terms. Let us consider the current mini-batch $\mathcal{B} \subseteq \mathcal{D}$, where $\mathcal{D}$ is the given training dataset. Let $\mathcal{C}_{\mathcal{B}}$ be the set of classes present in $\mathcal{B}$, i.e., $\mathcal{C}_{\mathcal{B}} = \{y \in \{1, 2 \cdots, C\} : (x, y) \in \mathcal{B}\}$. Let $\mathcal{C}_c^{\mathcal{B}}$ be the examples of class $c$ in $\mathcal{B}$, i.e., $\mathcal{C}_c^{\mathcal{B}} = \{x : (x, y) \in \mathcal{B}, y = c\}$.

To ensure intra-class compactness, we use the discriminative loss proposed in De Brabandere et al. (2017). Note that in the latter article, this loss is used in the different context of image segmentation. This loss is written

$$\mathcal{L}_{\text{compact}}(\Theta; \mathcal{B}) = \frac{1}{|\mathcal{C}_{\mathcal{B}}|} \times \sum_{c \in \mathcal{C}_{\mathcal{B}}} \frac{1}{|\mathcal{C}_c^{\mathcal{B}}|} \sum_{x \in \mathcal{C}_c^{\mathcal{B}}} [\|m_c - q(x)\| - \delta_v]_+^2 \ , \tag{8}$$

where $q(x) = f_\theta(x)$ (recall from Section 3.1, as we work in the feature space $\mathcal{F}$) and $m_c$ is the centroid of the class $c$. We will discuss how to compute these centroids in Section 4.4. Here, $\|.\|$ is the Euclidean norm, $[q]_+ = \max(0, q)$. This function is zero when $\|m_c - q\| < \delta_v$. Hence, this function enforces that the distance of each point to its centroid is smaller than $\delta_v$ (here, the subscript $v$ in $\delta_v$ denotes the *variance* of the class). Notice that this function only pushes the distance to be smaller than $\delta_v$ and not to be zero. Hence, we avoid the phenomenon of feature collapse.

To have a better inter-class separability, we build a loss function enforcing large margin between classes and the decision boundaries. A naive strategy would be to maximize distance of each sample to all the decision boundaries. However, this is very costly and not really necessary. Instead, we propose to maximize the distance of each centroids to the decision boundaries. Indeed, we will give in Proposition 5.2 a lower bound for the class margins. The margin loss is defined as follows,

$$\mathcal{L}_{\text{margin}}(\Theta; \mathcal{B}) = \frac{1}{|\mathcal{C}_{\mathcal{B}}|} \times \sum_{c \in \mathcal{C}_{\mathcal{B}}} \max_{i \neq c} \left[ \delta_d + d(m_c, \mathcal{P}_{ci}) \, \text{sign}(z(m_c)_i - z(m_c)_c) \right]_+ \ , \tag{9}$$

where we recall that $z(m_c) = W m_c + b \in \mathbb{R}^C$ and $\mathcal{P}_{ci}$ is defined in Eq.(6). This function is inspired by the work of Elsayed et al. (2018) [2]. Intuitively, when the centroid $m_c$ is on the right side of the decision boundary, $z(m_c)_i - z(m_c)_c < 0$. Hence, in this case we minimize $[\delta_d - d(m_c, \mathcal{P}_{ci})]_+$ and consequently $d(m_c, \mathcal{P}_{ci})$ is encouraged to be larger than $\delta_d$ (here, the subscript $d$ in $\delta_d$ denotes the *distance* from the centroid to the boundaries). In contrast, if $m_c$ is on the wrong side of the decision boundary, then we minimize $[\delta_d + d(m_c, \mathcal{P}_{ci})]_+$. This enforces $m_c$ to pass to the right side. Hence, this loss is only deactivated if the centroid is on the right side w.r.t all the decision boundaries and its distance to the decision boundaries are larger than $\delta_d$. Moreover, notice that we opt for the aggregation operation $\max_{i \neq c}$ instead of $\text{mean}_{i \neq c}$.

---

[2]Note that the work of Elsayed et al. (2018) focuses on maximizing an approximated Euclidean margin to the decision boundaries and does not impose constraints on intra-class compactness. As a result, their method maximizes the distance of each individual sample to the decision boundaries. In contrast, our approach maximizes the distance between class centroids and the decision boundaries, which significantly reduces computational cost. This is because we have already the intra-class compactness, which ensures that individual samples remain close to their corresponding class centroids.

Indeed, it may happen that some pairs of class are easier to separate than others. With *mean* aggregation, loss can be minimized by focusing only on easy pairs and ignoring difficult pairs. In contrast, with aggregation max, we enforce the neural networks to focus on difficult pairs. As such, it can learn more useful features to increase discriminative power. Notice that the distance of $m_c$ to the hyperplane $\mathcal{P}_{ci}$, the decision boundary of class pair $(c, i)$, can be computed explicitly as,

$$d(m_c, \mathcal{P}_{ci}) = \frac{|\langle W_c - W_i, m_c \rangle + (b_c - b_i)|}{\|W_c - W_i\|}. \tag{10}$$

Our loss function encourages each centroids to be far away from the decision boundaries. However, there are no guarantee that the decision boundaries lead to closed cells [3]. The resulted centroids could be pushed far away. Hence, to address this problem, we add a regularization term as proposed in De Brabandere et al. (2017),

$$\mathcal{L}_{\text{reg}}(\Theta; \mathcal{B}) = \frac{1}{|\mathcal{C}_{\mathcal{B}}|} \sum_{c \in \mathcal{C}_{\mathcal{B}}} \|m_c\| . \tag{11}$$

Note that this regularization term is optional, meaning that we can set $\gamma_{\text{reg}} = 0$.

We also note that we use the squared loss for $\mathcal{L}_{\text{compact}}$, while a non-squared version is used for $\mathcal{L}_{\text{margin}}$. The rationale behind this design choice is provided in Appendix A.1.

### 4.2 Aligning noisy data with clean data

In the section above, we have introduced the constraint to boost the discriminativeness of the features on clean data. Now, we need to introduce an additional constraint to ensure that the model produces more stable features under input perturbations. For this, let noised($x$) denote the noisy version of $x$ (after being injected with noise). We then introduce the following loss function,

$$\mathcal{L}_{\text{noisy}}(\Theta; \mathcal{B}) = \frac{1}{|\mathcal{C}_{\mathcal{B}}|} \times \sum_{c \in \mathcal{C}_{\mathcal{B}}} \frac{1}{|\mathcal{C}_c^{\mathcal{B}}|} \sum_{x \in \mathcal{C}_c^{\mathcal{B}}} [\|m_c - \widetilde{q}(x)\| - \delta_v]_+^2 , \tag{12}$$

where $\widetilde{q}$ is the corresponding feature of the perturbed input noised($x$), i.e., $\widetilde{q}(x) = f_\theta(\text{noised}(x))$.

We notice that the final constraint is similar to the intra-class compactness loss applied to the clean data. This constraint encourages the features of noisy samples to lie on the same hypersphere as those of clean samples from the same class. Unlike standard methods that align each noisy sample directly with its clean counterpart, our approach performs feature alignment only at the class level. This more flexible design helps maintain the expressiveness of the learned features.

### 4.3 Our loss

With all the components presented above, we come up with the total constraint as follows

$$\mathcal{L}_{\text{ours}} = \mathcal{L}_{\text{clean}} + \lambda \mathcal{L}_{\text{noisy}} , \tag{13}$$

where we fix $\lambda = 1$ for the sake of simplicity.

### 4.4 Computing class centroid: partial momentum strategy

In our loss presented above, the class centroids appear in different constraints (Eqs. (8), (9), (11) and (12)). We now discuss how to compute and update such class centroids during training. Let us now consider a

---

[3]By a *closed cell*, we mean the region in feature space bounded by the decision boundaries separating a given class from all others. In the ideal case, these boundaries enclose a finite region corresponding to the domain of that class. In practice, however, depending on the parameter configuration of the softmax classifier, the decision boundaries may fail to form a fully closed region.

class $c$. To compute the centroid of this class, we use only the clean data. This is because during training, the features of corrupted data can be very far away from those of clean data, or even fall in the wrong class. Consequently, this can make the centroid updated in the wrong direction. There are 2 straightforward ways:

- **Naive way.** Using all the sample of the considered class in the current mini-batch: $m_c^t := m_c^{\text{current}}(\Theta; \mathcal{B}) = \frac{1}{|\mathcal{C}_c^{\mathcal{B}}|} \sum_{x \in \mathcal{C}_c^{\mathcal{B}}} q(x)$.

- **Using momentum.** $m_c^t := m_c^{\text{momentum}} = \gamma \cdot m_c^{t-1} + (1 - \gamma) \cdot m_c^{\text{current}}(\Theta; \mathcal{B})$, where $\gamma$ is chosen to be close to 1, such as 0.9. Note that $m_c^{t-1}$ from the last batch is used as a fixed quantity here (no gradient).

One major advantage of using momentum is stability. Recall that in modern machine learning, very small mini-batches are now common to achieve speed-ups at the cost of noisy gradient steps. Thus it can happen that the centroid of each class fluctuates too much from one batch to another. In such case, we do not have a stable direction to that centroid. As $\mathcal{L}_{\text{compact}}$ aims to push each point to its corresponding centroid, the optimization becomes less effective. Hence, the use of momentum allows us to avoid this problem. However, using momentum makes the gradient much smaller when updating the model parameters. More precisely, we have following proposition:

**Proposition 4.1** $\nabla_\theta \mathcal{L}_{\text{margin}}^{\text{moment}} = (1 - \gamma) \cdot \nabla_\theta \mathcal{L}_{\text{margin}}^{\text{naive}}$. *Here, $\mathcal{L}_{\text{margin}}^{\text{naive}}$ and $\mathcal{L}_{\text{margin}}^{\text{moment}}$ are the margin losses computed using the centroids updated based on naive way and momentum way, respectively.*

**Proof 4.1** *See Appendix C.*

This proposition shows that using centroid with or without momentum gives the same gradient direction. Nevertheless, with momentum the very small shrinking scaling factor $1 - \gamma$ appears.

Further, this small gradient is multiplied by a small learning rate ( typically in the range $[10^{-5}, 10^{-2}]$). So, on the one hand, the parameter updating in the momentum method is extremely small (or even get completely canceled out by the computer rounding limit or *machine epsilon*). On the other hand, as discussed previously, using momentum allows more stability. To overcome the gradient drawback but to conserve the stability benefit, we combine the naive and momentum ways. We come up with a strategy named *partial momentum*. This strategy uses momentum for the compactness loss and naive way for the margin loss, respectively. Doing so, we have stable centroids. So that, each point is pushed in a stable direction. But at the same time, the centroids are kept *consistent* (*consistent* here means that centroids receive sufficiently large gradients to be back-propagated to the set of parameters $\theta$ of $f_\theta$, preventing them from becoming outdated during training).

## 5 Some properties of our loss function

### 5.1 Properties of intra-class compactness and inter-class separability

In this section, we investigate the properties of compactness and separability of the loss function. Furthermore, we discuss the impact of the hyper-parameters $\delta_v$ in $\mathcal{L}_{\text{compact}}$ and $\delta_d$ in $\mathcal{L}_{\text{margin}}$. This gives us a guideline on the choice of these hyper-parameters. We recall that $\mathcal{D}$ denotes the training dataset and $\mathcal{C}_c^{\mathcal{D}} = \{x : (x, y) \in \mathcal{D}, \ y = c\}$.

**Proposition 5.1** *If $\mathcal{L}_{\text{compact}}(\Theta; \mathcal{D}) = 0$, then the dispersion of all classes (see Definition 3.1) is at most $2\delta_v$.*

**Proof 5.1** *See Appendix B.1.*

This last proposition shows that the hinge center loss ensures the intra-compactness property of each class.

**Definition 5.1 (Class margin)** *Let us define the margin of a given class $c$ as the smallest Euclidean distance of samples in this class to its closest decision boundary, i.e.*

$$\text{margin}(c) := \min_{x \in \mathcal{C}_c^{\mathcal{D}}} \left( \min_{i \neq c} d(q(x), \mathcal{P}_{ci}) \right) \ .$$

**Proposition 5.2** *Assume that $\mathcal{L}_{\text{compact}}(\Theta; \mathcal{D}) = \mathcal{L}_{\text{margin}}(\Theta; \mathcal{D}) = 0$. Then,*

1. *If $\delta_d > \delta_v$, then the margin of all classes is at least $\delta_d - \delta_v$.*

2. *If $\delta_d > 2\delta_v$, then in the feature space $\mathcal{F}$, the distances between any points in the same class are smaller than the distances between any points from different classes.*

**Proof 5.2** *See Appendix B.2.*

Hence, if we aim to obtain class margin at least $\varepsilon$, then we can set $\delta_d = \delta_v + \varepsilon$. Furthermore, this proposition provides a guideline for the choice of $\delta_v$ and $\delta_d$. We are aiming for a representation with not only a large inter-class margin, but also one in which the distances between points in the same class are smaller than the distances between points in different classes. This is particularly useful for problems where samples in each class are too diverse whereas samples from different classes are too similar.

## 5.2 Intra-class compactness or inter-class separability constraint alone does not suffice

In order to boost class compactness, one can minimize the distance of each point to the corresponding class centroid. As presented in Section 2, this approach is proposed in Wen et al. (2016) using a center loss. Boosting the class margin is proposed by Elsayed et al. (2018). However, we shall prove that applying only one of these two approaches, the model can be encouraged to evolve in a direction that does not change the model prediction.

Indeed, they can lead to solutions that allow for enforcing class compactness or class margin but where the predictions of the model remain unchanged – and so does the generalization capability. In such cases, the model is not encouraged to learn more discriminative features. This also suggests that the use of both compactness and margin is necessary. More formally, let us recall that $\mathcal{N}$ (with the parameters $\Theta$) denotes the softmax model, as defined in Section 3.1. We suppose that the layer prior to the feature space $\mathcal{F}$ is a standard image convolutional layer or fully-connected one, that can be followed or not followed by the non-linearity *ReLU*. Suppose that the current parameters of $\mathcal{N}$ is $\Theta$ (denoted by $\mathcal{N}_\Theta$).

**Proposition 5.3** *For $\nu > 0$, there exists a map, depending on $\nu$, $\mathcal{T}_\nu$, such that $\tilde{\Theta} = \mathcal{T}_\nu(\Theta)$ satisfies $\mathcal{N}_{\tilde{\Theta}}(x) = \mathcal{N}_\Theta(x)$ for all $x \in \mathcal{X}$. Furthermore,*

1. *If $\mathcal{N}_\Theta$ is such that $\min_c \text{margin}(c) = d_{\text{margin}} > 0$ (see Definition 5.1), then, for the new model $\mathcal{N}_{\tilde{\Theta}}$, $\min_c \text{ margin}(c) = \nu d_{\text{margin}}$.*

2. *If $\mathcal{N}_\Theta$ is such that $0 < \max_c \text{dispersion}(c) = d_{\text{dispersion}}$ (see Definition 3.1), then, for the new model $\mathcal{N}_{\tilde{\Theta}}$, $\max_c \text{dispersion}(c) = \nu d_{\text{dispersion}}$.*

3. *$\mathcal{T}_\nu(\Theta) - \Theta = (\nu - 1)u$, where $u$ is a vector depending only on $\Theta$.*

**Proof 5.3** *See Appendix D.*

**Corollary 5.1** *$\forall \nu_1, \nu_2 > 1$, $\mathcal{T}_{\nu_1}(\Theta) - \Theta$ and $\mathcal{T}_{1/\nu_2}(\Theta) - \Theta$ are vectors of opposite directions. In particular, as $\nu_1$ or $\nu_2$ increase, the movement of parameters happens in opposite directions (illustrated in Fig. 5). That is, using this family of transformations, the class margin expansion and the class dispersion shrinkage cannot happen at the same time.*

Suppose that we are given a network $\mathcal{N}_\Theta$ with certain class margin and dispersion – and that was trained with softmax loss for example. We note that the statements (1) and (2) in Proposition 5.3 hold for all $\nu > 0$. This indicates that, by choosing $\nu < 1$ or $\nu > 1$, we can explicitly perform a transformation to obtain new parameters of $\mathcal{N}$ to increase the class margin or decrease the class dispersion — to satisfy the constraints imposed by either center loss or margin loss alone, respectively — without changing the predictions of $\mathcal{N}$.

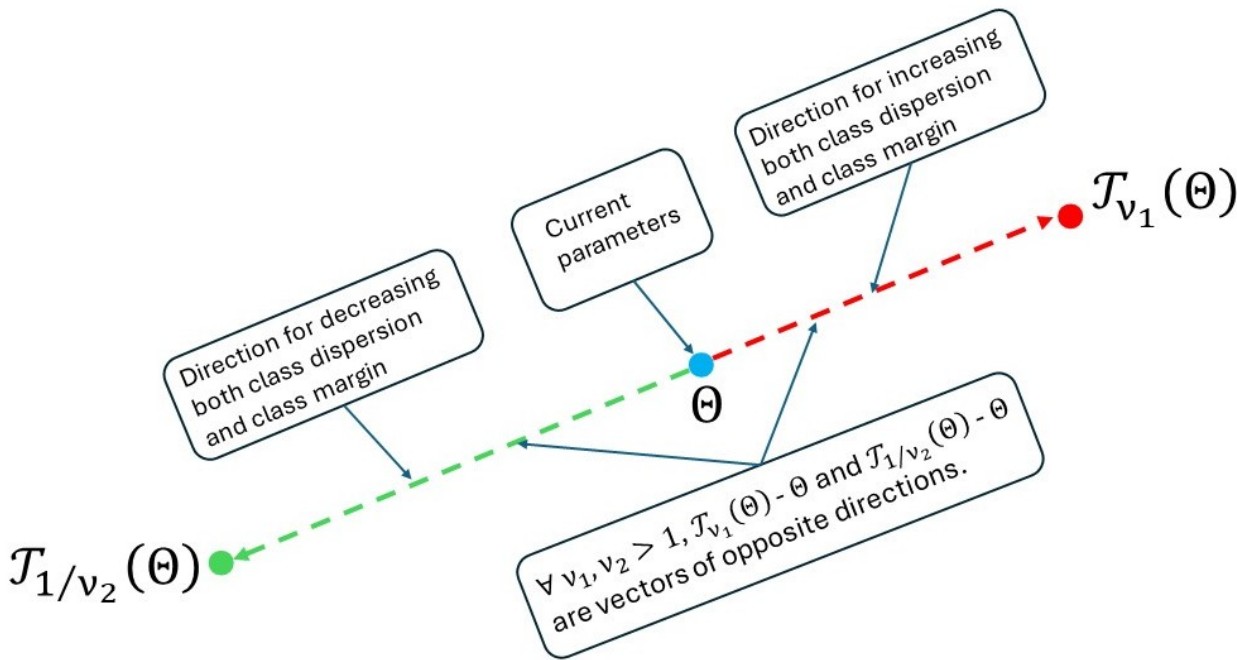

Figure 5: For any $\nu_1, \nu_2 > 1$, the vectors $\mathcal{T}_{\nu_1}(\Theta) - \Theta$ and $\mathcal{T}_{1/\nu_2}(\Theta) - \Theta$ point in opposite directions. Moving along either direction preserves model predictions but increases intra-class dispersion or decreases the inter-class margin, thereby violating the joint optimization enforced by our constraints. Therefore, our loss avoids such solutions $\mathcal{T}_\nu$, which do not improve generalization.

That is, the model is not really encouraged to learn more discriminative features and the generalization of the model is not improved. However, Corollary 5.1 tells that we cannot both expand the class margin and shrink the class dispersion simultaneously. This is because $\Theta$ is moved in two exactly opposite directions for these two objectives. In other words, moving along either of the two directions defined by $\mathcal{T}_\nu$ would increase the intra-class dispersion or decrease the inter-class margin, thereby violating the joint optimization enforced by our constraints. Consequently, our loss avoids such solutions $\mathcal{T}_\nu$, which do not lead to improved generalization.

## 6 Theoretical insights on our method

In this section, we present theoretical insights into our method. In Section 6.1, we analyze how our approach affects the curvature of the softmax loss in the input space. Since our method also enforces that the inputs are projected onto the hypersphere corresponding to their class, Section 6.2 investigates the generalization behavior of this constraint, based on how well it is empirically satisfied on the training set. Lastly, as our method is applied only at the penultimate layer (i.e., in the feature space $\mathcal{F}$), we analyze in Section 6.3 its potential impact on intermediate layers.

### 6.1 Impact of our method on the curvature of the softmax loss in the input space

When analyzing model stability with respect to input perturbations, a common approach is to examine the curvature of a standard loss function in the input space. Typical loss functions include the mean squared error for regression and the softmax loss for multiclass classification. Following this line of reasoning, in this section we analyze how our method effectively reduces the curvature of the softmax loss in the input space (that we called *input loss curvature*). Conversely, we also demonstrate that a reduction in this curvature leads to improved model stability under Gaussian input perturbation. Note that prior works (e.g., Moosavi-

Dezfooli et al. (2019)) have also shown experimentally that reducing the input loss curvature leads to models that are more robust to input perturbations.

**Background: input loss curvature and its relation with the Hessian matrix.** We first recall some background for studying the curvature of a loss function at an input point $x \in \mathbb{R}^k$ via its Hessian matrix (assume that the input domain $\mathcal{X} \subseteq \mathbb{R}^k$). Consider the softmax loss function (Eq. (4)):

$$l(x) = l(x, y) = -\log \sigma \left(W f_\theta(x) + b\right)_y .$$

Notice that, we aim at studying the curvature with respect to the input $x$, so we drop $y$ in the annotation for brevity, and we write $l(x)$. Assume that $l$ is twice continuously differentiable (with respect to $x$). The **Hessian matrix** $H(x)$ is defined as the matrix of the second-order partial derivatives:

$$H(x) = \nabla^2 l(x) = \begin{bmatrix} \frac{\partial^2 l(x)}{\partial x_1^2} & \cdots & \frac{\partial^2 l(x)}{\partial x_1 \partial x_k} \\ \vdots & \ddots & \vdots \\ \frac{\partial^2 l(x)}{\partial x_k \partial x_1} & \cdots & \frac{\partial^2 l(x)}{\partial x_k^2} \end{bmatrix} .$$

This can be interpreted as the Jacobian of the gradient vector: $H(x) = \frac{d}{dx} \nabla l(x)$. If we move in a direction $v \in \mathbb{R}^k$, we consider the path $x(t) = x + tv$. We notice that

$$\lim_{t \to 0} \frac{\nabla l(x + tv) - \nabla l(x)}{t} = \left. \frac{d}{dt} \nabla l(x + tv) \right|_{t=0} = H(x)v .$$

Hence, if $v$ is a unit vector ($\|v\| = 1$), $\|H(x)v\|$ describes the change in the gradient **per unit length** (or **gradient change rate**) at the point $x$, as we move in direction $v$. This can be regarded as the curvature along the direction $v$. Consider an eigenvector $v$ of $H$ ($\|v\| = 1$), with corresponding eigenvalue $\lambda$, then we have $\|H(x)v\| = \|\lambda v\| = |\lambda|$. Therefore, the set of absolute eigenvalue magnitudes $\{|\lambda_i|\}_{i=1}^k$ of $H(x)$ provides a representation of the geometric (unsigned) curvature of $l$ at point $x$. For instance, a way to quantify the overall curvature at $x$ is computing $\sum_i \lambda_i^2$ or $\sum_i |\lambda_i|$.

**Theoretical insights in the impact of our method on the softmax loss curvature.** Equipped with the background above, we now introduce our theoretical results. For a perturbation $\varepsilon$ around $x \in \mathcal{X}$, we assume the following approximation

$$l(x + \varepsilon) \approx l(x) + \nabla l(x)^T \varepsilon + \frac{1}{2} \varepsilon^t H(x) \varepsilon . \tag{14}$$

Consider an input $x$ from class $y$, with feature representation $q(x) = f_\theta(x)$ in the feature space $\mathcal{F}$. We use $f_\theta(x)$ instead of $q(x)$ to emphasize its dependence on the model. For $\delta > 0$, let $\mathcal{C}(f_\theta(x), \delta)$ denote the hypersphere of radius $\delta$ centered at $f_\theta(x)$. Under the compactness loss $\mathcal{L}_{\text{compact}}$ (Eq. (10)) for clean inputs and the alignment loss $\mathcal{L}_{\text{align}}$ (Eq. (12)) for noisy inputs, a well-trained model ensures that both clean and noisy features of the same class lie within a hypersphere of radius $\delta_v$ around the class centroid. Hence, their distance is at most $2\delta_v$. Therefore, under a Gaussian perturbation $\varepsilon$, we expect

$$f_\theta(x + \varepsilon) \in \mathcal{C}(f_\theta(x), \delta), \quad \text{with } \delta = 2\delta_v,$$

with high probability. This observation is the key link between input loss curvature and model stability under Gaussian perturbations. More formally, we define stability under Gaussian perturbations as follows.

**Definition 6.1 (Stability under Gaussian perturbations)** *Consider a Gaussian noise $\varepsilon \sim \mathcal{N}(\mathbf{0}, \sigma^2 \mathbf{I})$. For a given $\delta > 0$, let $\eta = \mathbb{P}_\varepsilon \left( f_\theta(x + \varepsilon) \in \mathcal{C}(f_\theta(x), \delta) \right), \quad l_{out} = \mathbb{E}_\varepsilon \left[ l(x + \varepsilon) \mid f_\theta(x + \varepsilon) \notin \mathcal{C}(f_\theta(x), \delta) \right].$*

- ***Feature stability.** We say that $f_\theta$ produces* stable features *w.r.t. $\varepsilon$ (at $x$) if $\eta$ is large for small $\delta$.*

- **Loss stability.** *We say that $f_\theta$ produces* stable loss *w.r.t. $\varepsilon$ (at $x$) if $|l_{out} - l(x)|$ is small. That is, even when features fall outside $\mathcal{C}(f_\theta(x), \delta)$, the resulting loss does not deviate much from the clean loss $l(x)$ (in expectation).*

In the extreme case of **feature stability**, where $\eta = 1$ for $\delta = 0$, $f_\theta$ produces features that are almost surely invariant under the noise $\varepsilon$. In this case, we will show that the input loss curvature is zero. Conversely, we will also show that a low input loss curvature encourages at least **feature stability** or **loss stability**.

Assuming that the logit values are bounded within $[-K_{\max}, K_{\max}]$ for all inputs, the following theorem formalizes the connection between input loss curvature and model stability under additive Gaussian noise.

**Theorem 6.1 (Upper-bound and lower-bound of the input loss curvature)** *Assume that $\varepsilon$ follows the distribution $\mathcal{N}(\mathbf{0}, \sigma^2 \mathbf{I})$, and let $\eta = \mathbb{P}_\varepsilon \left( f_\theta(x + \varepsilon) \in \mathcal{C}(f(x)_\theta, \delta) \right)$ (we assume that $\eta > 0$). Then,*

1. *Denoting the set of eigenvalues of $H(x)$ by $\{\lambda_i\}_{i=1}^k$ and $\|W\|_{2,\infty} = \max_j \|W_j\|$, we have*

$$\sum_i \lambda_i^2 \leq \frac{8}{\sigma^4} \left( \eta \|W\|_{2,\infty}^2 \delta^2 + 4(1 - \eta) \cdot K_{\max}^2 \right) \ . \tag{15}$$

2. *Let $l_{out} = \mathbb{E}_\varepsilon \left[ l(x + \varepsilon) \mid f_\theta(x + \varepsilon) \notin \mathcal{C}(f_\theta(x), \delta) \right]$, we have*

$$2\sum_i \lambda_i^2 + \left( \sum_i |\lambda_i| \right)^2 \geq \frac{4}{\sigma^4} \left( (1 - \eta) \cdot (l_{out} - l(x))^2 - \sigma^2 \|\nabla l(x)\|^2 \right) \ . \tag{16}$$

**Proof 6.1** *See Appendix E.*

**Remark 6.1** *Theorem 6.1 contains two statements that address complementary regimes. Statement (1) considers a setting in which feature stability holds, and we aim to show that improved feature stability leads to smaller input loss curvature. In contrast, Statement (2) focuses on another regime where feature or loss stability has **not yet** been achieved (i.e., when $(1 - \eta) \cdot (l_{out} - l(x))^2$ is sufficiently large). We aim to understand what happens if we can somehow reduce the input loss curvature in this regime.*

**Statement (1).** In Eq. (15), the left-hand side (LHS) $\sum_i \lambda_i^2$ represents the input loss curvature. In the right-hand side (RHS), if $\eta$ increases to 1 then the second term decreases to 0. An obvious way to increase $\eta$ is by increasing $\delta$ (there is a higher chance that $f_\theta(x + \varepsilon) \in \mathcal{C}(f_\theta(x), \delta)$). However, by doing so, we also increase the first term of the RHS (as it depends on $\delta$). Therefore, one way to minimize the upper bound on the curvature is to train the model such that $f_\theta(x + \varepsilon) \in \mathcal{C}(f_\theta(x), \delta)$ with high probability $\eta$, for a small $\delta$. This also suggests that choosing $\delta$ close to zero is not a good choice. Indeed, in this case it is difficult to train a model such that $\eta$ is large. With our intra-compactness constraint, we push the features of the clean and noisy inputs ($f_\theta(x)$ and $f_\theta(x + \varepsilon)$) to lie within the hypersphere $\mathcal{C}(m_y, \delta_v)$ of the same class. Hence, if the model is well trained and taking $\delta = 2\delta_v$, with high probability we have $f_\theta(x + \varepsilon) \in \mathcal{C}(f_\theta(x), \delta)$. So, this effectively helps to decrease the input loss curvature. This will be demonstrated in our experiments (see Section 7.4).

**Statement (2).** As mentioned in Remark 6.1, we focus on the regime in which feature (or loss) stability has not been achieved, namely when $(1 - \eta)(l_{out} - l(x))^2$ is sufficiently large. Under this regime, consider Eq. (16). If the gradient of the loss at $x$ is sufficiently small and noting that $\sigma$ is typically small, we have $\sigma^2 \|\nabla l(x)\|^2 \approx 0$. Consequently, this term can be neglected compared to $(1 - \eta)(l_{out} - l(x))^2$. In this case,

$$2\sum_i \lambda_i^2 + \left( \sum_i |\lambda_i| \right)^2 \gtrsim \frac{4(1 - \eta)}{\sigma^4} \cdot (l_{out} - l(x))^2 \ .$$

This inequality suggests that, for a fixed $\delta$, reducing the input loss curvature (i.e., the Hessian eigenvalues) either increases $\eta$ or encourages a smaller value of $|l_{out} - l(x)|$. This indicates that reducing the curvature

of the loss with respect to the input encourages at least **feature stability** or **loss stability**. Besides, recall that our loss function includes a constraint that explicitly increases the margin between clean sample features and the classification decision boundaries. As a result, when noisy features remain close to their clean counterparts, the classifier is more likely to make consistent predictions under perturbations. This highlights the complementary role of the margin-based constraint in promoting classification stability.

As a concluding remark, this last theorem reveals a strong connection between the input loss curvature and the model stability.

## 6.2 Generalization behavior of the compactness constraint for mapping features onto a hypersphere

Recall that our method enforces inputs to be projected onto the hypersphere associated with their class via the intra-class compactness constraint $\mathcal{L}_{\text{compact}}$ in Eq. (8). In this section, we fix a particular class and focus on the compactness constraint. We assume that $\mathcal{L}_{\text{compact}}$ vanishes on the training set; that is, for all training samples $x$, we have $\|f(x) - m\| \leq \delta_v$. We now examine whether compactness is preserved on the test data. More concretely, for a given $r > 0$, let $\mathcal{C}(m, r)$ denote the hypersphere centered at $m$ with radius $r$. We then study the probability that a test point $x$ is projected inside $\mathcal{C}(m, r)$. In short, we investigate how well the compactness observed on the training set generalizes to the test set.

To this end, we first recall the notion of margin loss introduced in Mohri (2018).

**Definition 6.2 (Margin loss function)** *For any $\rho > 0$, the $\rho$-margin loss function is defined as*

$$\Phi_\rho(\tau) = \begin{cases} 1 & \text{if } \tau \leq 0, \\ 1 - \tau/\rho & \text{if } 0 \leq \tau \leq \rho, \\ 0 & \text{if } \tau \geq \rho \,. \end{cases} \tag{17}$$

Now, for a given $r > 0$ and $0 < \rho < r^2$, define $h(x) := r^2 - \|f(x) - m\|^2$. Then,

$$\Phi_\rho(h(x)) = \begin{cases} 1 & \text{if } \|f(x) - m\| \geq r, \\ 1 - (r^2 - \|f(x) - m\|^2)/\rho & \text{if } \sqrt{r^2 - \rho} \leq \|f(x) - m\| \leq r, \\ 0 & \text{if } \|f(x) - m\| \leq \sqrt{r^2 - \rho} \,. \end{cases} \tag{18}$$

**Remark 6.2** *When $\rho \to 0$, the function in Eq. (18) penalizes inputs $x$ that are projected outside $\mathcal{C}(m, r)$ (i.e., $\|f(x) - m\| \geq r$). This allows us to quantify the* projection error. *For larger $\rho$, the function also penalizes points lying inside $\mathcal{C}(m, r)$ but within a margin $\rho$ of the boundary (i.e., $\sqrt{r^2 - \rho} \leq \|f(x) - m\| \leq r$). Thus, the parameter $\rho$ can be interpreted as a confidence margin.*

In our method, by enforcing the model to satisfy the compactness constraint on the training set, we expect the compactness property to generalize to the test set. *Is this a reasonable objective?* To address this question, we quantify the mapping error, i.e., the probability that a point is projected outside $\mathcal{C}(m, r)$ for a given $r > 0$.

Recall that $\mathcal{F}$ denotes the feature space, and let $\mathcal{M}(\mathcal{X}, \mathcal{F})$ be the set of all measurable functions from $\mathcal{X}$ to $\mathcal{F}$. For parameters $R, r, \Lambda > 0$, we consider the following function class:

$$H = \left\{ h(\cdot) = r^2 - \|f(\cdot) - m\|^2 \; : \; \|m\| \leq R, \; f \in \mathcal{M}(\mathcal{X}, \mathcal{F}), \; \sup_{x \in \mathcal{X}} \|f(x)\| \leq \Lambda \right\} . ^{[4]}$$

This class allows both the feature mapping $f$ and the hypersphere center $m$ to vary, subject to boundedness assumptions. Such assumptions are natural in light of the regularization term in Eq. (11). The next theorem provides an empirical generalization bound on the mapping error. Let $S$ denote a sample consisting of $N > 0$ i.i.d. copies of $X$.

---

[4]The boundedness parameters essentially refer to the class centroids and the outputs of $f$. Specifically, the regularization term in Eq. (11) encourages the learned features to remain within a bounded range. Intuitively, if the feature space were unbounded, generalization would be severely impaired, as features could occupy arbitrarily large regions of the space.

**Theorem 6.2** *For any $\delta > 0$, with probability at least $1 - \delta$ over the draw of an i.i.d. sample $S$ of size $N$, the following holds for all $h \in H$:*

$$\mathbb{P}(h(X) < 0) \leq \widehat{R}'_{S,\rho}(h) + \frac{2}{\rho}\left(\Lambda^2 + 2R\Lambda + \frac{R^2}{\sqrt{N}}\right) + 3\sqrt{\frac{\log\frac{2}{\delta}}{2N}} \ . \tag{19}$$

*Here, $\widehat{R}'_{S,\rho}(h) = \frac{1}{N}\sum_{i=1}^{N}\Phi_\rho(h(x_i))$ denotes the empirical error on $S$.*

**Proof 6.2** *See Appendix F.1.*

**Remarks.** If an input $x$ is mapped inside the hypersphere in the feature space, then $h(x) > 0$. Thus, $\mathbb{P}(h(X) < 0)$ measures the mapping error, i.e., the probability that inputs are mapped outside the hypersphere.

From Eq. (18) and Remark 6.2, we note that $\widehat{R}'_{S,\rho}(h)$ penalizes training examples $x$ such that $\sqrt{r^2 - \rho} \leq \|f(x) - m\|$. In particular, if $\delta_v \leq \sqrt{r^2 - \rho}$, then $\widehat{R}'_{S,\rho}(h) = 0$. Besides, observe that the upper bound in Eq. (19) decreases as the number of training examples $N$ increases. This is natural, since a larger training set provides better coverage of the underlying input distribution. Consequently, if the model is well trained (i.e., the empirical loss on the training set is small), it is more likely to generalize well to unseen data.

### 6.3 Impact on the features of intermediate layers

Our loss is applied only to the features at the penultimate layer. This naturally raises an important question: *does it also influence the representations learned in the earlier, intermediate layers?* In this section, we provide theoretical insights into this question. In short, Theorem 6.3 shows that enforcing inter-class separability at the penultimate layer allows us to derive a lower bound on the inter-class separability of the features in intermediate layers.

Let $X \in \mathcal{X}$ and $Y \in \mathcal{Y}$ denote the random variables (r.v.), modeling respectively the input and the label. Let $\mathcal{D}_X$ and $\mathcal{D}_Y$ be the distributions of $X$ and $Y$, respectively. Assume that $f_\theta$ is composed of $L$ layers. For a given layer $l \in \{1, 2, \cdots, L\}$, let $Q^l$ denote the feature r.v. of this layer. That is, the distribution $\mathcal{D}_{Q^l}$ of $Q^l$ is an induced distribution of $\mathcal{D}_X$. Given $y \in \mathcal{Y}$, let $\mathcal{D}_{Q^l|y}$ be the induced distribution from the conditional input distribution $\mathcal{D}(X|Y = y)$.

We recall that our loss operates at the outputs of $f_\theta$, i.e., the feature space of layer $L$. This aims to enforce features of different classes to be contained in different hyper-spheres. Hence, in some sense, it pushes away $\mathcal{D}_{Q^L|y}$ from $\mathcal{D}_{Q^L|y'}$, for any $(y, y') \in \mathcal{Y}^2$ ($y \neq y'$). In this way, the features of different classes can be more separable.

Now, let us consider an arbitrary intermediate layer $l \leq L$ before the penultimate layer. The question is: *For all $(y, y') \in \mathcal{Y}^2$ ($y \neq y'$), does our loss $\mathcal{L}_{clean}$ make $\mathcal{D}_{Q^l|y}$ more separated from $\mathcal{D}_{Q^l|y'}$?* To measure the divergence between two distributions $\mathcal{D}_1$ and $\mathcal{D}_2$, we use the Jensen–Shannon divergence (JSD), denoted by $D_{\mathrm{JS}}(\mathcal{D}_1 \,\|\, \mathcal{D}_2)$. For completeness, we refer the reader to Appendix G for formal definitions.

Now, let us consider a trained model with our method. Recall that the compactness loss encourages the features in $\mathcal{F}$ of each class $y$ to be contained in a hyper-sphere of radius $\delta_v$ and centered at $m_y$, denoted by $\mathcal{C}(m_y, \delta_v)$. Thus, it is natural to make the following margin assumption

(H)      There exists a $\tau$ ($1/2 \leq \tau \leq 1$) such that for all $y \in \mathcal{Y}$, we have

$$\mathcal{D}_{Q^L|y}(\mathcal{C}(m_y, \delta_v)) := \mathbb{P}(Q^L \in \mathcal{C}(m_y, \delta_v)|Y = y) \geq \tau.$$

Under this assumption, the probability that each point is projected in the right hyper-sphere is at least $\tau \geq 1/2$. This is a reasonable assumption if the model is well trained. Under the assumption (H), we have the following theorem.

**Theorem 6.3** *Let $y, y' \in \mathcal{Y}$ such that $y \neq y'$. For any intermediate layer $l \leq L$ before the penultimate layer, we have*

$$D_{JS}(\mathcal{D}_{Q^l|y} \,\|\, \mathcal{D}_{Q^l|y'}) \geq (2\tau - 1)^2/2 \ . \tag{20}$$

**Proof 6.3** *See Appendix G.*

**Remark 6.3** *Note that our goal is to derive a model-agnostic lower bound that holds universally, without relying on architectural details. Nevertheless, under additional assumptions on the layer type or network structure, it may be possible to obtain sharper, layer-dependent bounds.*

This theorem gives a lower bound for the JSD between the conditional feature distributions of intermediate layers. If the model is well trained, $\tau$ gets larger, so does $D_{JS}(\mathcal{D}_{Q^l|y} \,||\, \mathcal{D}_{Q^l|y'})$. That is, in the intermediate layers, features of different classes become more separated from each other. The intuition behind this is rather simple. Indeed, let us argue using contradiction reasoning in an informal manner. Suppose for example that $\mathcal{D}_{Q^l|y}$ is very similar to $\mathcal{D}_{Q^l|y'}$ and that they share a large part of their supports. Then, the features of this layer are propagated until the feature space $\mathcal{F}$ by the same mapping – which is the set of layers between layer $l$ and the penultimate layer. Hence, the two conditional distributions $\mathcal{D}_{Q^L|y}$ and $\mathcal{D}_{Q^L|y'}$ in $\mathcal{F}$ should be also similar. This contradicts the condition imposed by our loss, where features of different classes should be contained in distinct hyper-spheres. Hence, $\mathcal{D}_{Q^l|y}$ should be distinct from $\mathcal{D}_{Q^l|y'}$ so that these constraints on the penultimate layer are satisfied. Theorem 6.3 formalizes this idea. Indeed, this matches the results observed in our experiments (see Section 7.5).

# 7 Experiments

In this section, we conduct experiments to evaluate the robustness of our method in comparison with baseline approaches (Section 7.2). In addition, we aim to verify whether our method can preserve model performance on clean data (relative to training without noise injection), while simultaneously improving robustness against input perturbations (Section 7.2). We further investigate in Section 7.3 the relationship between robustness and the curvature of the input loss surface, providing empirical evidence that supports our theoretical result in Theorem 6.1. Moreover, we demonstrate that our method effectively reduces input loss curvature (Section 7.4). Finally, we present qualitative results showing that, when our method is applied at the penultimate layer, the intermediate feature representations also become more discriminative (Section 7.5), thereby supporting the theoretical findings in Theorem 6.3.

## 7.1 General experimental details

Here, we provide some general experimental details; all additional information can be found in Appendix H. In our experiments, we use 3 different datasets.

- **CIFAR-10 dataset.** CIFAR-10 (Krizhevsky et al., 2009) consists of 50,000 training images and 10,000 test images across 10 classes, including both vehicles and animals. These are $32 \times 32$ color images.

- **Street View House Numbers (SVHN) dataset.** SVHN (Netzer et al., 2011) contains 73,257 training images and 26,032 test images of house numbers captured from Google Street View. These are also $32 \times 32$ RGB images.

- **Road Condition Image Dataset.** We construct a custom dataset tailored to our task, comprising 1,897 road images collected from both publicly available sources and our own road image recording campaign. All the images are resized to $400 \times 500 \times 3$. The dataset is divided into three classes: *dry* (460 images), *wet/watered* (460 images), and *icy/snowy* (977 images). To ensure diversity, we deliberately select images from a wide range of scenes within each class. Some examples from our dataset are depicted in Fig. 6.

**Neural network models.** For the two datasets CIFAR10 and SVHN, we use ResNet18 (He et al., 2016) as backbone, followed by a fully connected layers prior to softmax layer [5]. For our own dataset, we test on the model MobileNetV3 (Howard et al., 2019), as this model is sufficiently lightweight for real-time application on vehicles. Note that the entire model is trained end-to-end, with no layers frozen.

---

[5]More precisely, the backbone is followed by a fully connected layer with output dimension 128 preceding the softmax layer, which has 10 output classes. See Appendix H for details.

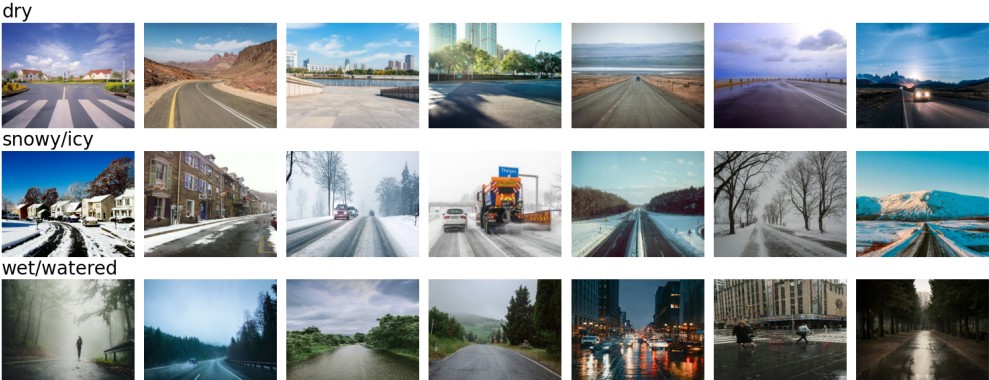

dry

snowy/icy

wet/watered

Figure 6: Some examples from our custom road image dataset, including 3 categories.

**Noisy test accuracy.** For each type of input perturbation, we perform 10 independent runs and report the average and standard deviation (std) of the test accuracy under noise.

**Baseline methods.** We compare our approach with the following baseline methods.

- **Normal training.** We train models with the standard softmax loss on clean training data (with standard data augmentation, see Appendix H).

- **Training only on noisy data.** We apply Gaussian noise (on top of data augmentation) to clean data for training model with the standard softmax loss .

- **Training on both clean and noisy data.** Besides noisy data, we also keep the clean data. Then the model is trained on both clean and noisy data with the standard softmax loss.

- **Stability training.** This method was first proposed in Zheng et al. (2016). It introduces a stability regularization term in addition to the standard softmax loss on the clean data. The stability regularization encourages the feature representations of clean and noisy versions of the same input to be close, by minimizing their distance. For a fair comparison, we apply this constraint to the same feature space $\mathcal{F}$ as used in our method, so we minimize $\|f_\theta(x) - f_\theta(x + \varepsilon)\|$ during training, where $\varepsilon$ is Gaussian noise in our experiments.

Note that in our method, the proposed constraints are applied alongside the standard softmax loss on clean data, as this helps produce more separable features (as discussed in Section 3). For simplicity, we fix $\alpha = \beta = 1$ and $\gamma_{\text{reg}} = 10^{-3}$ for all the experiments.

**Calculation of the input loss curvature.** We notice that the curvature of the softmax loss function at a point $x$ can be calculated as $\sum_i \lambda_i^2 = \mathbb{E}_\epsilon \left[ \varepsilon^T H(x)^2 \varepsilon \right] = \|H(x)\varepsilon\|^2$, where $\varepsilon \sim \mathcal{N}(0, \mathbf{I})$. Moreover, for any $\varepsilon$, by definition of the Hessian, $H(x)\varepsilon = \frac{d}{dt}H(x + t\varepsilon)\big|_{t=0} = \lim_{t \to 0} \frac{\nabla l(x+t\varepsilon) - \nabla l(x)}{t}$. Hence, we can estimate the input loss curvature without calculating the second derivatives w.r.t. inputs as follows:

$$\Lambda(x) := \sum_i \lambda_i^2 \approx \frac{1}{K} \sum_{j=1}^K \frac{1}{t^2} \left\| \nabla l(x + t\varepsilon_j) - \nabla l(x) \right\|^2 , \tag{21}$$

where $\varepsilon_j$ are i.i.d. samples from $\mathcal{N}(0, \mathbf{I})$. In our experiments, we fix $t = 10^{-2}$ and $K = 20$. Preliminary results show that $K = 20$ already provides stable estimates, with small variance across different approximations.

### 7.2 Model performance under input perturbations

**Experiments on standard datasets CIFAR-10 and SVHN.** For CIFAR-10, we inject additive Gaussian noise with std $= 0.06$ during training for all methods. After training, the models are evaluated under varying

levels of additive Gaussian noise, as shown in Table 1. We follow the same procedure for SVHN, but use a higher noise level of std = 0.15 during training, since SVHN appears more robust to small perturbations compared to CIFAR-10. The results for CIFAR-10 and SVHN are reported in Tables 1 and 2, respectively.

Table 1: Accuracy on noisy test set at different Gaussian noise level for CIFAR10. Note that for training, we set the standard deviation (std) of the Gaussian noise to 0.06.

| Additive Gaussian noise level (std) | Clean | 2/255 | 4/255 | 8/255 | 16/255 | 20/255 |
|---|---|---|---|---|---|---|
| Normal training | **92.73** | $92.63 \pm 0.09$ | $91.95 \pm 0.09$ | $87.73 \pm 0.19$ | $64.93 \pm 0.18$ | $50.32 \pm 0.29$ |
| Training only on noisy images | 89.64 | $89.65 \pm 0.06$ | $89.70 \pm 0.07$ | $89.99 \pm 0.08$ | $89.81 \pm 0.14$ | $87.918 \pm 0.16$ |
| Training on both clean and noisy images | 91.48 | $91.45 \pm 0.04$ | $91.41 \pm 0.07$ | $91.15 \pm 0.11$ | $90.06 \pm 0.14$ | $89.04 \pm 0.21$ |
| Stability training | 91.13 | $91.16 \pm 0.04$ | $91.24 \pm 0.09$ | $91.26 \pm 0.08$ | $89.12 \pm 0.15$ | $84.62 \pm 0.21$ |
| **Ours** | 92.67 | **92.70** $\pm 0.05$ | **92.68** $\pm 0.07$ | **92.61** $\pm 0.07$ | **91.15** $\pm 0.20$ | **89.42** $\pm 0.11$ |

Table 2: Accuracy on noisy test set at different Gaussian noise level for SVHN. Note that for training, we set the standard deviation (std) of the Gaussian noise to 0.15.

| Additive Gaussian noise level (std) | Clean | 5/255 | 10/255 | 20/255 | 36/255 | 42/255 |
|---|---|---|---|---|---|---|
| Normal training | 95.69 | $95.41 \pm 0.06$ | $94.32 \pm 0.08$ | $86.85 \pm 0.13$ | $66.36 \pm 0.21$ | $58.87 \pm 0.16$ |
| Training only on noisy images | 93.42 | $93.39 \pm 0.03$ | $93.37 \pm 0.04$ | $93.02 \pm 0.07$ | $91.64 \pm 0.15$ | $90.39 \pm 0.11$ |
| Training on both clean and noisy images | 95.34 | $95.25 \pm 0.02$ | $94.96 \pm 0.07$ | $93.94 \pm 0.07$ | $91.62 \pm 0.09$ | $90.43 \pm 0.08$ |
| Stability training | 96.01 | $95.93 \pm 0.03$ | $95.63 \pm 0.07$ | $94.59 \pm 0.06$ | $91.45 \pm 0.14$ | $89.71 \pm 0.07$ |
| **Ours** | **96.13** | **96.06** $\pm 0.03$ | **95.91** $\pm 0.05$ | **95.17** $\pm 0.07$ | **92.47** $\pm 0.09$ | **90.86** $\pm 0.08$ |

From Tables 1 and 2, we observe that training solely on noisy data, or on both clean and noisy data, tends to degrade performance on uncorrupted (clean) or mildly corrupted test samples (i.e., perturbations smaller than 4/255 for CIFAR10 and 10/255 for SVHN). In contrast, when the noise level is sufficiently high (i.e., $\geq 8/255$ for CIFAR10 and $\geq 20/255$ for SVHN), training on noisy data can actually outperform standard training. This suggests that noisy training helps the model fit to the noisy distribution, but does not necessarily promote learning of well-generalized features that perform well on clean inputs.

On CIFAR-10, stability training notably reduces clean-data accuracy, whereas our method maintains performance comparable to normal training. On SVHN, our method even improves accuracy on clean data. With small perturbations (2/255 for CIFAR-10 and 5/255 for SVHN), the performance gain over normal training is limited. However, as the noise level increases, our method consistently outperforms both normal training and alternative approaches. Specifically, it yields improvements of up to nearly 30% over the normal training baseline at the highest noise level on both datasets. Furthermore, our method systematically outperforms the stability training counterpart, demonstrating its effectiveness.

**The model extrapolates stability beyond the noise level seen during training.** For each dataset (Tables 1 and 2), we additionally evaluate the trained models under noise levels higher than those used during training, namely 20/255 for CIFAR-10 and 42/255 for SVHN. At these magnitudes, the injected noise already significantly alters the semantic content of the inputs, making evaluation at substantially higher noise levels less meaningful. Despite this, we observe that the model trained with our method retains strong performance when tested beyond the training noise regime. This behavior indicates that the learned stability properties generalize beyond the specific noise levels encountered during training, providing evidence of enhanced robustness.

**Qualitative results.** We use the t-SNE technique to visualize the learned feature representations of CIFAR-10 in two-dimensional space, as shown in Fig. 1. The visualization reveals that under normal training, features are not well separated when evaluated on noisy data (right plot of Fig. 1a). When trained on both clean and noisy data (Fig. 1b), feature separation improves on noisy inputs, but the structure of clean data becomes less distinct compared to normal training. This trade-off leads to a performance drop on clean inputs. In contrast, our method enhances feature separability on both clean and noisy data (Fig. 1c). These qualitative observations are consistent with the quantitative results, further demonstrating the effectiveness of our approach.

**Experiments on our custom road image dataset.** We randomly split the dataset into 80% for training and 20% for testing. During training, all methods are exposed to additive Gaussian noise with std $= 40/255$. Additional training details are provided in Appendix H. Once trained, the models are evaluated under various types of input perturbations to assess their robustness.

We evaluate model robustness under the following perturbations (illustrated in Fig. 7):

- **Additive Gaussian noise.** As in the CIFAR-10 and SVHN experiments, we apply varying noise levels (standard deviation), each evaluated over 10 independent runs to ensure statistical reliability.

- **Random occlusion.** We randomly mask 20 patches of size $70 \times 70$ per image. Results are averaged over 10 independent runs.

- **Downsampling-Upsampling (DU sampling).** In this method, we first downsample each image by a factor of 1/3 in both dimensions, then upsample it by a factor of 3 to restore the original size. This operation is deterministic, so we report results from a single run.

- **Random Stripe Masking.** We randomly mask 10 vertical stripes of thickness equal to 6.

- **Combination.** We also evaluate the combined effects of (i) random occlusion with Gaussian noise, (ii) random occlusion with DU sampling and (iii) DU sampling with random stripe masking.

**Results on additive Gaussian noise.** The results at different noise levels are reported in Table 3. Our method consistently outperforms the other methods. Notably, it slightly improves performance even on clean data. At the noise level $40/255$, our approach yields an improvement of more than 30% compared to the baseline of standard training. At a noise level of $60/255$, which exceeds the training noise level ($40/255$), the model trained with our method retains strong performance. When using For the other methods, we observe the same phenomenon as in CIFAR10 and SVHN.

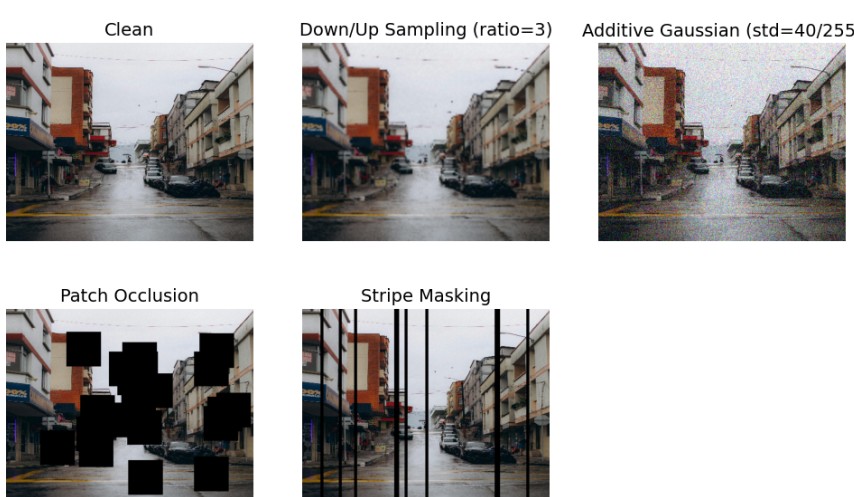

Figure 7: Examples of perturbations applied on the inputs at inference, including Gaussian noise, random occlusion, down/up sampling and stripe masking. In occlusion, we randomly mask 20 patches of size $70 \times 70$. In stripe masking, we randomly mask 10 stripes of thickness equal to 6.

**Results for other perturbations (Table 4).** We further evaluate the models under additional perturbations, including random occlusion, DU sampling and stripe masking, as previously described. For random occlusion, our method significantly outperforms the other methods. Notably, while stability training performs well under additive Gaussian noise, it yields poor results under random occlusion. When combining random occlusion with additive Gaussian noise, our method again surpasses the others. On DU sampling, our method is slightly outperformed by stability training. However, when DU sampling is combined with

Table 3: Accuracy on noisy test set at different Gaussian noise level for road data. During training, we set the standard deviation (std) of the Gaussian noise to 40/255.

| Additive Gaussian noise level (std) | clean | 10/255 | 30/255 | 40/255 | 60/255 |
|---|---|---|---|---|---|
| normal training (clean data) | 96.84 | $95.57 \pm 0.28$ | $75.42 \pm 1.08$ | $59.34 \pm 1.28$ | $43.81 \pm 1.02$ |
| training with noisy data | 91.58 | $92.66 \pm 0.41$ | $95.71 \pm 0.24$ | $94.95 \pm 0.62$ | $92.05 \pm 0.84$ |
| training with noisy and clean data | 96.31 | $96.36 \pm 0.35$ | $95.97 \pm 0.58$ | $95.39 \pm 0.73$ | $92.92 \pm 0.46$ |
| stability training | 96.05 | $96.31 \pm 0.23$ | $94.63 \pm 0.58$ | $92.92 \pm 0.80$ | $90.87 \pm 0.69$ |
| **ours** | **97.37** | **97.29** $\pm 0.17$ | **96.71** $\pm 0.49$ | **95.52** $\pm 0.64$ | **93.65** $\pm 0.49$ |

occlusion, our method clearly demonstrates superior performance over all other methods. This is also the case with random stripe masking. These experiments strongly support the conclusion that training with additive Gaussian noise alone with our method can enhance the model's robustness to a broader range of perturbations beyond Gaussian noise.

Table 4: Accuracy on noisy test set for road data under input perturbations. The standard deviation (std) of Gaussian noise is 40/255. For random occlusion, we randomly mask 20 patches of size $70 \times 70$. In stripe masking, we randomly mask 10 stripes of thickness equal to 6. DU Sampling stands for Downsampling-Upsampling, where we down-sample image resolution by 1/3 and then up-sample by the factor of 3 to recover the initial size.

| Perturbation type | Normal training | Training only on noisy images | Training on both clean and noisy images | Stability training | **Ours** |
|---|---|---|---|---|---|
| Occlusion | $89.89 \pm 0.49$ | $80.21 \pm 1.02$ | $87.92 \pm 0.68$ | $80.99 \pm 1.19$ | **94.05** $\pm 0.75$ |
| Occlusion + Gaussian noise | $27.68 \pm 0.30$ | $74.00 \pm 1.23$ | $78.60 \pm 1.48$ | $54.07 \pm 1.32$ | **82.73** $\pm 1.45$ |
| DU Sampling | 92.36 | 86.58 | 91.84 | **96.05** | 95.79 |
| DU Sampling + Occlusion | $77.58 \pm 1.09$ | $78.86 \pm 1.35$ | $81.34 \pm 0.71$ | $74.47 \pm 1.02$ | **88.05** $\pm 0.55$ |
| Stripe masking | $84.39 \pm 1.30$ | $76.21 \pm 1.23$ | $88.00 \pm 0.65$ | $86.63 \pm 1.16$ | **90.42** $\pm 0.62$ |
| DU Sampling + Stripe masking | $69.52 \pm 1.62$ | $70.13 \pm 1.33$ | $77.78 \pm 1.06$ | $82.44 \pm 0.99$ | **87.99** $\pm 1.53$ |

### 7.3 Correlation between input loss curvature and model stability under input perturbations

The results of Theorem 6.1 suggest a strong correlation between the input loss curvature with the model performance under input perturbations. In this section, we perform experiments to observe this curvature-stability relation in practice, where we use uniform and Gaussian noises.

**Noisy test accuracy as a function of retained low-curvature samples.** Consider the clean test set $\mathcal{D}_{\text{test}} = \{(x_i, y_i)\}_{i=1}^N$. For each sample $x_i$, we estimate the loss curvature $\Lambda(x_i)$ using the procedure in Eq. (21), and collect the results into $\Lambda = \{\Lambda(x_i)\}_{i=1}^N$. We then evaluate noisy test accuracy by retaining different proportions of low-curvature samples. Specifically, for $p \in [0, 1]$, we compute the $p$-th quantile of $\Lambda$, denoted by $\text{Quantile}(\Lambda, p)$. We define the subset

$$\mathcal{D}_{\text{test}}(p) := \{(x_i, y_i) \in \mathcal{D}_{\text{test}} : \Lambda(x_i) \leq \text{Quantile}(\Lambda, p)\}.$$

The average noisy accuracy on this subset is computed as

$$\text{acc}(\mathcal{D}_{\text{test}}(p)) := \frac{1}{|\mathcal{D}_{\text{test}}(p)|} \sum_{(x_i, y_i) \in \mathcal{D}_{\text{test}}(p)} \mathbb{1}_{\{\widehat{y}(x_i + \varepsilon_i) = y_i\}},$$

where $\widehat{y}(x_i + \varepsilon_i)$ denotes the model prediction under noisy input $x_i + \varepsilon_i$, and we consider both uniform and Gaussian noise.

By varying $p$, we obtain the results in Fig. 8. For both standard training and our proposed method, the noisy test accuracy decreases as a larger portion of samples is retained. Equivalently, restricting evaluation to fewer (lower-curvature) samples yields higher robustness to noise. This behavior indicates that robustness is strongly correlated with low-curvature inputs, consistent with the theoretical result in Theorem 6.1.

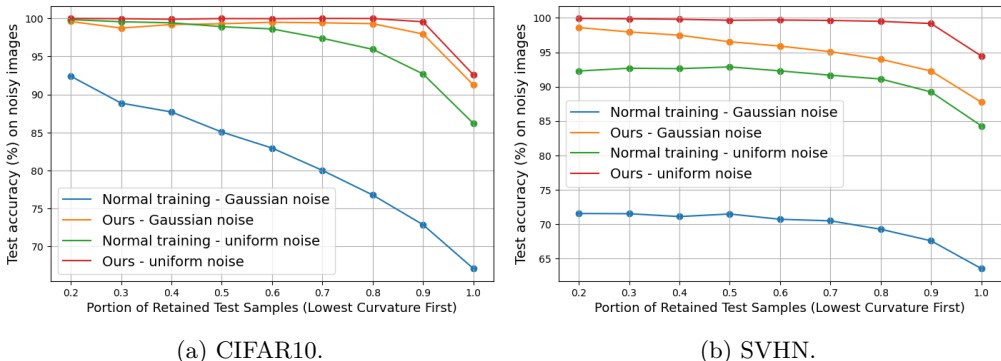

(a) CIFAR10.          (b) SVHN.

Figure 8: Test accuracy on noisy images for different retained portion of test samples with lowest curvature. We conduct experiments using both models trained with our method and normal training. For CIFAR-10, we set the standard deviation of the Gaussian noise and the amplitude of the uniform noise to 0.06. For SVHN, we fix both values to 0.15.

**Further analysis of the robustness–curvature relationship.** To further investigate the connection between robustness and input loss curvature, we conduct the following experiment. For each test sample $(x_i, y_i) \in \mathcal{D}_{\text{test}}$, we record how many times the model produces a correct prediction over 10 independent random noise perturbations:

$$\text{count}(i) = \sum_{j=1}^{10} \mathbb{1}_{\{\widehat{y}(x_i+\varepsilon_j)=y_i\}},$$

where $\widehat{y}(x_i + \varepsilon_j)$ denotes the model prediction under noisy input $x_i + \varepsilon_j$.

Based on this count, we partition the test set into groups according to the number of correct predictions. Specifically, for each $k \in \{0, 1, \ldots, 10\}$, we define

$$G(k) := \{\, i : \text{count}(i) = k \,\}.$$

We then compute the average input loss curvature (on clean inputs) for each group:

$$\bar{\Lambda}(k) := \frac{1}{|G(k)|} \sum_{i \in G(k)} \Lambda(x_i).$$

Finally, we plot $k$ versus $\bar{\Lambda}(k)$ to analyze how robustness correlates with input loss curvature. The results for both uniform and Gaussian noise on CIFAR-10 and SVHN are shown in Fig. 9. We observe a strong negative correlation between the number of correct predictions and the average input loss curvature, indicated by a high Pearson correlation coefficient. This provides additional evidence that lower input loss curvature is associated with greater robustness to input noise.

### 7.4 Impact of our method on the input loss curvature

Theoretical results from Theorem 6.1 indicate that training a model to be robust to Gaussian noise should lead to a reduction in input loss curvature. Our method enforces that features from clean and noisy inputs lie on the same hypersphere for each class, promoting feature stability under Gaussian perturbations. As a result, the input loss curvature is expected to decrease.

Given that the noisy test accuracy of the best-performing method is around 90%, we select the 90% of test samples with the lowest input loss curvatures for each method and compute their curvature values. Fig. 10

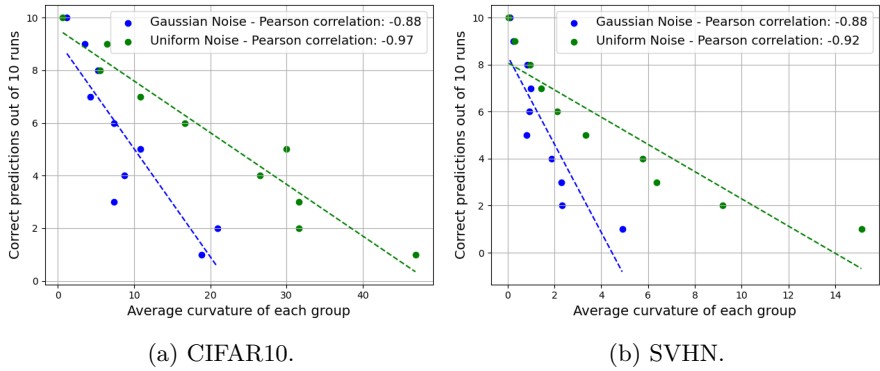

(a) CIFAR10.           (b) SVHN.

Figure 9: Average input loss curvature for groups of test samples, partitioned by the number of correct predictions over 10 random noise injections. For CIFAR-10, we set the standard deviation of the Gaussian noise and the amplitude of the uniform noise to 0.06. For SVHN, we fix both values to 0.15.

shows boxplots of the curvature distributions for each training method. Our method significantly reduces input loss curvature compared to standard training. While training on both clean and noisy data also lowers curvature, it is less effective than our approach.

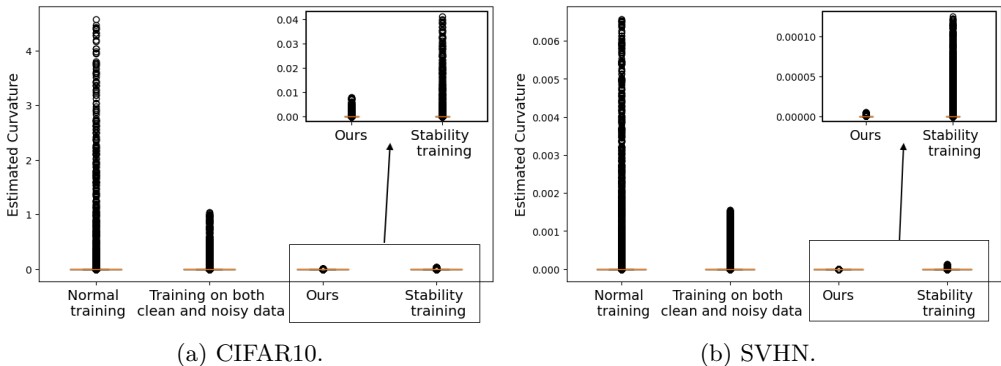

(a) CIFAR10.           (b) SVHN.

Figure 10: Boxplot of the top 90% lowest curvature values on the test set for each method. Since the accuracy of the best-performing method is around 90%, we visualize whether better performing method has lower curvatures among the top 90% lowest-curvature samples.

We also qualitatively examine the curvature of the input-space loss surface. For a fixed test example, we sample a grid of points in the input space along two directions: the gradient of the loss with respect to the input, and a random direction orthogonal to it. We then visualize the negative of the loss function over this 2D grid, as shown in Fig. 11. The results show that our method significantly smooths the loss landscape compared to the baselines. This qualitative observation is consistent with the quantitative results in Fig. 10, where our method notably reduces the input-space curvature of the loss.

### 7.5  Impact of our method on intermediate layers

In the experiments above, we use ResNet18 (He et al., 2016) as the backbone, followed by a fully connected layer prior to the softmax layer. Recall that our loss function is applied to the features of the penultimate layer (i.e., the layer immediately before the softmax). Thus, it is natural to ask whether the proposed loss influences only the final fully connected layers, or whether it also affects the backbone feature extractor. Figure 1 visualizes the features at the penultimate layer and shows improved class separability under our method. However, one could argue that this improvement is limited to the penultimate layer and does not reflect changes in the backbone representations. While Section 6.3 provides theoretical insights into how our constraint affects the intermediate layers of the network, we also investigate this question experimentally. To

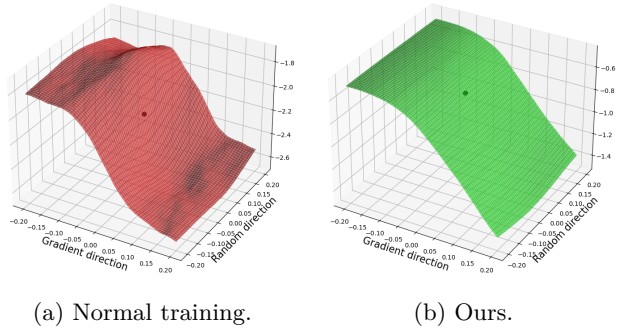

(a) Normal training. (b) Ours.

Figure 11: Illustration on 2D surface of the negative of the standard softmax loss at a given input of SVHN. Original input is marked by the black dot.

do so, we apply the same t-SNE visualization technique to the intermediate feature maps of the ResNet18 backbone. Specifically, ResNet18 consists of four main residual blocks, each producing outputs of dimension $H_i \times W_i \times C_i$ for $i = 1, \ldots, 4$, where $H_i$ and $W_i$ are the spatial dimensions and $C_i$ is the number of output channels for block $i$. For each block output, we apply *Global Average Pooling* over the spatial dimensions to obtain a feature vector of size $C_i$. We then apply t-SNE to these pooled features, as done for the penultimate layer. The resulting visualizations for the clean and noisy CIFAR-10 test set are shown in Figures 12 and 13, respectively.

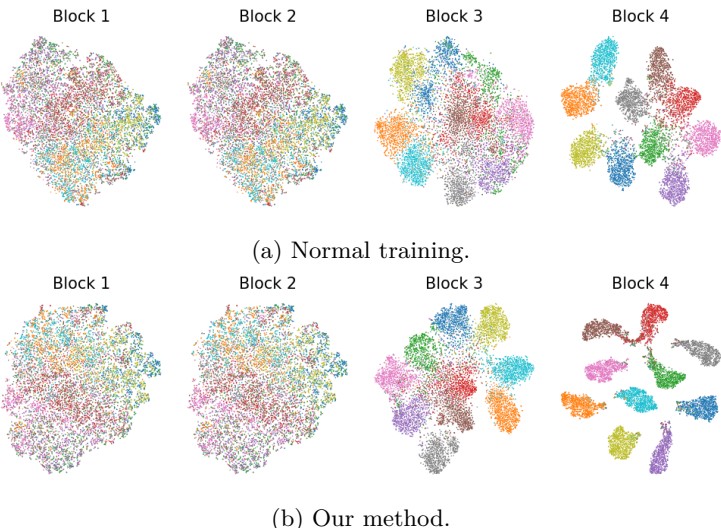

Figure 12: t-SNE visualization for features on clean CIFAR10 test set of different intermediate layers of the models trained with normal approach and our approach.

We observe that the features extracted from the first two blocks do not form distinctive clusters, which is expected since these early layers primarily capture low-level patterns rather than semantic class information. Starting from block 3, however, our loss function begins to encourage the formation of more clearly separated clusters compared to standard training, as shown in Figures 12 and 13. This effect becomes especially pronounced in block 4 for both clean and noisy inputs. In particular, the t-SNE plots for the noisy data (Figure 13) show a striking difference between our method and the standard baseline. While standard training fails to produce well-separated class clusters under noise, our method significantly enhances feature discriminability, even in the presence of perturbations. These findings indicate that our loss function not only improves representations at the penultimate layer but also promotes the learning of more structured and discriminative features in intermediate layers of the backbone. This empirical evidence aligns with our

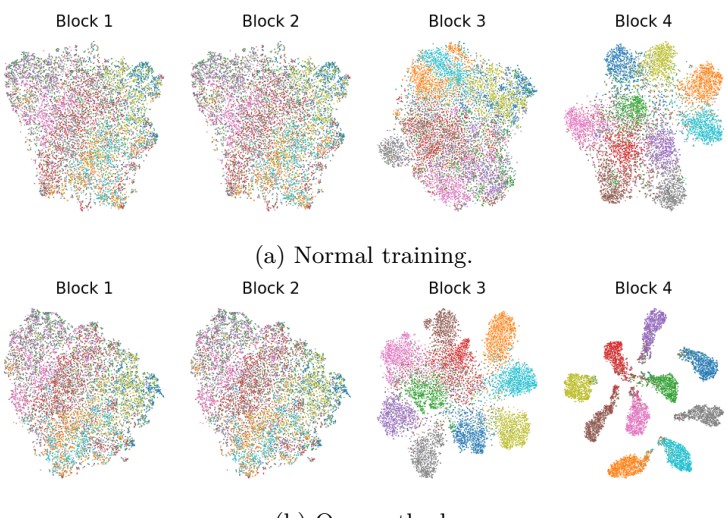

(a) Normal training.

(b) Our method.

Figure 13: t-SNE visualization for features on noisy CIFAR10 test set of different intermediate layers of the models trained with normal approach and our approach (Gaussian noise with std equal to 0.06).

theoretical results presented in Theorem 6.3, further supporting the claim that our approach affects the model's internal representations in a meaningful and robust manner.

## 7.6 Ablation study: complementarity of the constraints in our method

To investigate the complementarity of the proposed constraints, we conduct an ablation study. Note that the constraint imposed on noisy data plays a role similar to enforcing intra-class compactness on clean data. Therefore, to isolate the individual contributions of inter-class separability and intra-class compactness, we restrict our ablation study to constraints applied on clean data only. This avoids a potential form of *implicit enforcement* of compactness through noisy-data constraints, which could otherwise obscure the interpretation of the results.

We first fix the regularization weight to $\gamma_{\text{reg}} = 10^{-3}$ and consider three configurations: (i) $\alpha = \beta = 1$, (ii) $\alpha = 0$, $\beta = 1$, and (iii) $\alpha = 1$, $\beta = 0$, where $\alpha$ and $\beta$ control the strengths of the intra-class compactness and inter-class separability constraints, respectively. This set of experiments aims to assess the individual and joint effects of the two constraints. Next, to evaluate the role of regularization, we fix $\alpha = \beta = 1$ and compare three settings: $\gamma_{\text{reg}} = 0$, $\gamma_{\text{reg}} = 10^{-3}$ and $\gamma_{\text{reg}} = 1$.

The results are summarized in Table 5. Several observations can be drawn from these experiments. First, combining both compactness and separability constraints consistently outperforms using either constraint alone, highlighting their complementary roles. Second, when applied in isolation, the compactness constraint yields better performance than the separability constraint. This aligns with our earlier discussion: while the softmax loss inherently encourages inter-class separability, it does not explicitly enforce intra-class compactness, making the compactness constraint particularly beneficial. Finally, when $\alpha = \beta = 1$, incorporating regularization leads to improved performance compared to the unregularized case ($\gamma_{\text{reg}} = 0$).

From a practical perspective, if extensive tuning is undesirable, a small regularization value such as $\gamma_{\text{reg}} = 10^{-3}$ provides a robust choice, as it performs well across different ablation settings.

Table 5: Ablation Study

| Hyper-params | $\alpha = 1$, $\beta = 0$, $\gamma_{\text{reg}} = 0.001$ | $\alpha = 0$, $\beta = 1$, $\gamma_{\text{reg}} = 0.001$ | $\alpha = 1$, $\beta = 1$, $\gamma_{\text{reg}} = 0.001$ | $\alpha = 1$, $\beta = 1$, $\gamma_{\text{reg}} = 0$ | $\alpha = 1$, $\beta = 1$, $\gamma_{\text{reg}} = 1$ |
|---|---|---|---|---|---|
| CIFAR10 | 93.43 | 92.25 | 94.01 | 93.66 | 93.81 |
| SVHN | 96.15 | 95.98 | 96.64 | 96.55 | 96.67 |

# 8 Conclusion

We presented a simple and general training framework that improves the robustness of deep neural networks to input perturbations while maintaining strong accuracy on clean data. Our method introduces a loss at the penultimate layer that is easy to integrate into existing models and training routines without requiring architectural changes or complex optimization procedures. This loss promotes intra-class compactness and inter-class separability of clean features, while also aligning noisy features at the class level—encouraging robustness without sacrificing expressiveness. From a theoretical perspective, we demonstrated that training with additive Gaussian noise implicitly regularizes the input-space curvature of the loss function by reducing the eigenvalues of its Hessian. This provides a principled explanation for the improved stability and robustness observed in practice. Importantly, we also show that this curvature reduction leads to more stable feature representations under noise, establishing a bidirectional connection between noise injection and input-space geometry. Empirically, our approach consistently improves performance on both clean and noisy data. Moreover, models trained with our method on Gaussian noise exhibit strong generalization to other types of corruptions, including occlusion and resolution degradation. This demonstrates robustness beyond the specific perturbation seen during training. Overall, our work offers a theoretically grounded, easy-to-adopt, and empirically validated approach for enhancing robustness in modern deep learning systems.

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

## Appendix A  More insights about our loss function

### A.1  To square the loss or not to square the loss

We can notice that each term under the sum operation in $\mathcal{L}_{\text{compact}}$ is squared, whereas this is not the case for $\mathcal{L}_{\text{margin}}$. In this section, we will discuss the rationale behind this design choice. We notice that terms under the sum operation of $\mathcal{L}_{\text{compact}}$ and $\mathcal{L}_{\text{margin}}$ can be written in a general form as

$$\phi(u) = \begin{cases} [\varepsilon\|u - u_{\text{ref}}\| + b]_+, & \text{if non-squared version} \\ ([\varepsilon\|u - u_{\text{ref}}\| + b]_+^2, & \text{if squared version} \end{cases}$$

where $\varepsilon$ is equal to 1 or $-1$ (the sign indicator) and $u_{\text{ref}}$ is the reference point. In the case of $\mathcal{L}_{\text{compact}}$, $\|u - u_{\text{ref}}\|$ is the distance from a generic point to its corresponding centroid denoted here by $u_{\text{ref}}$. In the case of $\mathcal{L}_{\text{margin}}$, $\|u - u_{\text{ref}}\|$ is the distance of a centroid to its projection on the closest boundary. For sake of simplicity, we ignore the sign indicator $\varepsilon$ before $\|u - u_{\text{ref}}\|$ as here this does not matter. Consider the case where $\phi(u)$ is still activated, i.e., $\|u - u_{\text{ref}}\| + b > 0$. In this case, we can show that:

$$\nabla_u \phi = \begin{cases} \frac{u - u_{\text{ref}}}{\|u - u_{\text{ref}}\|}, & \text{if non squared version} \\ 2(\|u - u_{\text{ref}}\| + b) \times \frac{u - u_{\text{ref}}}{\|u - u_{\text{ref}}\|}, & \text{if squared version} \end{cases},$$

which implies

$$\|\nabla_u \phi\| = \begin{cases} 1, & \text{if non squared version} \\ 2 \times \left|\|u - u_{\text{ref}}\| + b\right|, & \text{if squared version} \end{cases}.$$

Consider the regime where $u$ approaches $u_{\text{ref}}$. If not squared, the gradient remains of constant magnitude as long as it is still activated. In contrast, if the term is squared, then, for states close to be deactivated, $|\|u - u_{\text{ref}}\| + b|$ is close to 0. Hence, the gradient is very small. Thus, the magnitude of the update direction for $u$ becomes minimal when it is close to the deactivated state.

Following this discussion, on the one hand, squaring in $\mathcal{L}_{\text{compact}}$ gives a smoother and less stringent loss. For example, if there are very abnormal points, then this condition does not enforce completely the points to be in the hyper-sphere around their centroid. On the other hand, by not squaring for the margin loss, we enforce stronger conditions on the centroid until it attains the deactivated state. That is, its distance to the closest boundary is at least larger than $\delta_d$. This is important as the position of each centroid impact the distribution of the whole class. This insight justifies our proposed loss functions.

## Appendix B  Proofs of intra-class compactness and inter-class separability

### B.1  Proof of Proposition 5.1

**Proof B.1** *Let $q_1$ and $q_2$ be 2 arbitrary points (in feature space $\mathcal{F}$) in any class $c$. By triangle inequality, we have: $\|q_1 - q_2\| \le \|q_1 - m_c\| + \|m_c - q_2\| \le \delta_v + \delta_v = 2\delta_v$.*

### B.2  Proof of Proposition 5.2

**Proof of Proposition 5.2.1**

**Proof B.2** *Consider an arbitrary class $c$. Let $p$ be a point in this class (in feature space $\mathcal{F}$) and let $q$ be any point on the hyperplane $\mathcal{P}_{ci}$ for any $i \ne c$. Then, by triangle inequality, we have:*

$$\|m_c - q\| \le \|m_c - p\| + \|p - q\|.$$

*As $d(m_c, \mathcal{P}_{ci}) \ge \delta_d$, we have $\|m_c - q\| \ge \delta_d$, $\forall q \in \mathcal{P}_{ci}$. At the same time, we also have $\|m_c - p\| \le \delta_v$. Consequently, we obtain:*

$$\delta_d \le \|m_c - q\| \le \|m_c - p\| + \|p - q\| \le \delta_v + \|p - q\|.$$

*Hence, $\|p - q\| \geq \delta_d - \delta_v$, $\forall y \in \mathcal{P}_{ci}$. So, by definition, $d(p, \mathcal{P}_{ci}) \geq \delta_d - \delta_v$. This holds $\forall p \in \mathcal{C}_c$ and $\forall i \neq c$. Hence, $\mathrm{margin}(c) = \min_{p \in \mathcal{C}_c}\left(\min_{i \neq c} d(p, \mathcal{P}_{ci})\right) \geq \delta_d - \delta_v$. As we choose an arbitrary class c, this holds for all classes. The proposition is proved.*

**Proof of Proposition 5.2.2**

**Proof B.3** *Let $q_1$ and $q_2$ be 2 arbitrary points in any class same c. By Proposition 5.1, we have: $\|q_1 - q_2\| \leq 2\delta_v$. Now, let p and q by 2 arbitrary points in any two different classes i and j, respectively. It suffices to show that $\|p - q\| > 2\delta_v$ with $\delta_d > 2\delta_v$. Again, by triangle inequality, we have:*

$$\|m_i - m_j\| \leq \|m_i - p\| + \|p - m_j\| \leq \|m_i - p\| + (\|p - q\| + \|q - m_j\|).$$

*So, $\|p - q\| \geq \|m_i - m_j\| - (\|m_i - p\| + \|q - m_j\|) \geq \|m_i - m_j\| - 2\delta_v$. Now, let us consider $\|m_i - m_j\|$. By Cauchy-Schwart inequality, we have:*

$$\left|\left\langle m_i - m_j, \frac{W_i - W_j}{\|W_i - W_j\|}\right\rangle\right| \leq \|m_i - m_j\| \cdot \frac{\|W_i - W_j\|}{\|W_i - W_j\|} = \|m_i - m_j\|.$$

*Hence,*

$$\|m_i - m_j\| \geq \left|\frac{\langle W_i - W_j, m_i\rangle}{\|W_i - W_j\|} - \frac{\langle W_i - W_j, m_j\rangle}{\|W_i - W_j\|}\right|$$
$$= \left|\frac{\langle W_i - W_j, m_i\rangle + (b_i - b_j)}{\|W_i - W_j\|} - \frac{\langle W_i - W_j, m_j\rangle + (b_i - b_j)}{\|W_i - W_j\|}\right|$$

*As $m_i$ and $m_j$ are on 2 different sides of the hyperplane $\mathcal{P}_{ij}$ (which is the decision boundary), $\langle W_i - W_j, m_i\rangle + (b_i - b_j)$ is of opposite sign of $\langle W_i - W_j, m_j\rangle + (b_i - b_j)$. Hence,*

$$\left|\frac{\langle W_i - W_j, m_i\rangle + (b_i - b_j)}{\|W_i - W_j\|} - \frac{\langle W_i - W_j, m_j\rangle + (b_i - b_j)}{\|W_i - W_j\|}\right| = \left|\frac{\langle W_i - W_j, m_i\rangle + (b_i - b_j)}{\|W_i - W_j\|}\right| + \left|\frac{\langle W_i - W_j, m_j\rangle + (b_i - b_j)}{\|W_i - W_j\|}\right|.$$

*Consequently,*

$$\|m_i - m_j\| \geq \left|\frac{\langle W_i - W_j, m_i\rangle + (b_i - b_j)}{\|W_i - W_j\|}\right| + \left|\frac{\langle W_i - W_j, m_j\rangle + (b_i - b_j)}{\|W_i - W_j\|}\right|$$
$$= d(m_i, \mathcal{P}_{ij}) + d(m_j, \mathcal{P}_{ij}) \geq 2\delta_d.$$

*So, $\|p - q\| \geq \|m_j - m_j\| - 2\delta_v \geq 2\delta_d - 2\delta_v$. With $\delta_d > 2\delta_v$, we have $\|p - q\| > 2\delta_v$. So, $\|p - q\| > \|q_1 - q_2\|$.*

## Appendix C  Proof of Gradients in Proposition 4.1

**Proof C.1** *If $\mathcal{L}_{\mathrm{margin}} = 0$, then the gradient w.r.t. θ is 0, hence proposition is proved. Let us now consider the case where $\mathcal{L}_{\mathrm{margin}} > 0$ is still larger than 0. We can easily see that*

$$\nabla_\theta \mathcal{L}_{\mathrm{margin}} = \; sign((m_c)_i - (m_c)_c) \cdot \nabla_\theta d(m_c, \mathcal{P}_{cj}).$$

*Hence, if we show $\nabla_\theta d(m_c^{current}, \mathcal{P}_{ci}) = (1 - \gamma) \cdot \nabla_\theta d(m_c^{\mathrm{moment}}, \mathcal{P}_{ci})$, the proposition is proved. First, we recall that for two differentiable functions: $g_1 : \mathbb{R}^{d_1} \mapsto \mathbb{R}^{d_2}$ and $g_2 : \mathbb{R}^{d_2} \mapsto \mathbb{R}^{d_3}$, and if g denotes the composed function $g := g_2 \circ g_1$, then for $x \in \mathbb{R}^{d_1}$ we have the following chain rule for computing the Jacobian matrix:*

$$J_g(x) = J_{g_2}(g_1(x)) \times J_{g_1}(x).$$

*In the case where $d_3 = 1$, then we have:*

$$\nabla_x g = J_g(x)^T = J_{g_1}^T(x) \times J_{g_2}^T(g_1(x)) = J_{g_1}^T(x) \times \nabla_{g_1(x)} g_2.$$

*Now, recall that $d(m_c, \mathcal{P}_{ci}) = \frac{|\langle W_c - W_i, m_c \rangle + (b_c - b_i)|}{\|W_c - W_i\|}$. For sake of brevity, we consider the case where $\langle W_c - W_i, m_c \rangle + (b_c - b_i) \geq 0$ and the argument is the same for the case $\langle W_c - W_i, m_c \rangle + (b_c - b_i) < 0$. We have, $d(m_c, \mathcal{P}_{ci}) = \frac{\langle W_c - W_i, m_c \rangle + (b_c - b_i)}{\|W_c - W_i\|}$. Using the chain rule from Eq. (C.1) with $g_2(\cdot) = \frac{|\langle W_c - W_i, \cdot \rangle + (b_c - b_i)|}{\|W_c - W_i\|}$, we get:*

$$\nabla_\theta d(m_c^{current}, \mathcal{P}_{ci}) = J_{m_c}^T(\theta) \times \nabla_{m_c} g_2.$$

*On the one hand, we can easily show that $\nabla_{m_c} g_2$ is independent of $m_c$ and equals to $\frac{W_c - W_i}{\|W_c - W_i\|}$. So,*

$$\nabla_\theta d(m_c, \mathcal{P}_{ci}) = J_{m_c}^T(\theta) \times \frac{W_c - W_i}{\|W_c - W_i\|}.$$

*On the other hand, we have that $m_c^{\text{moment}} = \gamma \cdot m_i^{t-1} + (1 - \gamma) \cdot m_c^{current}$. Consequently,*

$$J_{m_c^{\text{moment}}}^T(\theta) = (1 - \gamma) \cdot J_{m_c^{current}}^T(\theta).$$

*Hence, $\nabla_\theta d(m_c^{\text{moment}}, \mathcal{P}_{ci}) = (1 - \gamma) \cdot \nabla_\theta d(m_c^{current}, \mathcal{P}_{ci})$. So,*

$$\nabla_\theta \mathcal{L}_{\text{margin}}^{\text{moment}} = (1 - \gamma) \cdot \nabla_\theta \mathcal{L}_{\text{margin}}^{\text{naive}}.$$

## Appendix D   Proof of Proposition 5.3

Recall that we work in the feature space where each point is denoted by $q$. Each $q$ itself is the output of its prior layer. As assumed in the theorem, this is a convolutional layer or fully connected layer, with or without a non-linearity *ReLU*. Convolutional or fully-connected layer can be written in a general matrix-operation form as,

$$q = \begin{cases} W^- q^- + b^-, & \text{if no non-linearity.} \\ \text{ReLU}(W^- q^- + b^-), & \text{if ReLU.} \end{cases}$$

Here, $W^-$ and $b^-$ are parameters of the prior layer, $q^-$ is the input of this layer. We denote this layer by $h_{(W^-, b^-)}$. We recall that $\text{ReLU}(a) = \max(0, a)$, $\forall a \in \mathbb{R}$. When *ReLU* is applied to a vector, we understand that *ReLU* is applied to each element of the vector.

**Remark D.1** *It is obvious that $h_{\nu(W^-, b^-)}(q^-) = \nu h_{(W^-, b^-)}(q^-)$, $\forall \nu \in \mathbb{R}$.*

We also recall that the softmax layer has parameters $(W, b)$ (see Eq. (1)). For ease of notation, we refer to $(W^-, b^-, b)$ as a flattened colunm-vector concatenating $W^-$, $b^-$ and $b$. Furthermore, we denote $\Theta = \begin{pmatrix} \Theta_I \\ (W^-, b^-, b) \end{pmatrix}$, i.e., $\Theta_I$ is the vector containing all the parameters of $\mathcal{N}$ except $(W^-, b^-, b)$. For an $\nu > 0$, we define $\mathcal{T}_\nu$ as

$$\mathcal{T}_\nu : \begin{pmatrix} \Theta_I \\ (W^-, b^-, b) \end{pmatrix} \mapsto \begin{pmatrix} \Theta_I \\ \nu(W^-, b^-, b) \end{pmatrix} := \begin{pmatrix} \tilde{\Theta}_I \\ (\tilde{W}^-, \tilde{b}^-, \tilde{b}) \end{pmatrix}.$$

In words, by applying $\mathcal{T}_\nu$ on $\Theta$, $(W^-, b^-, b)$ is multiplied by $\nu$, that is to say $(\tilde{W}^-, \tilde{b}^-, \tilde{b}) = \nu(W^-, b^-, b)$, and other parameters of $\Theta$ stay unchanged ($\tilde{\Theta}_I = \Theta_I$).

**1. First, we prove that after applying $\mathcal{T}_\nu$, the predictions of model do not change.** Consider an arbitrary input. By Remark D.1, we know that $q$ is transformed into $\tilde{q} = \nu q$ . Moreover, from Eq. (1), we have

$$\tilde{z} = W\tilde{q} + \tilde{b} = \tilde{z} = W\nu q + \nu b = \nu(Wq + b) = \nu z .$$

Now, by Remark 3.1, the outputs of the new model and the old model are $\arg\max_i \nu z_i$ and $\arg\max_i z_i$, respectively. As $\nu > 0$, $\arg\max_i \nu z_i = \arg\max_i z_i$. So, the new model has exactly the same prediction as the old one.

**2. Now, let us consider the class margin.** Recall that the distance from $q$ to the decision boundary $\mathcal{P}_{ij}$ (between class $i$ and $j$) is $d(q, \mathcal{P}_{ij}) = \frac{|\langle W_i - W_j, q\rangle + (b_i - b_j)|}{\|W_i - W_j\|}$. So, the distance after the transformation is

$$
\begin{aligned}
d(\tilde{q}, \tilde{\mathcal{P}}_{ij}) &= \frac{|\langle W_i - W_j, \tilde{q}\rangle + (\tilde{b}_i - \tilde{b}_j)|}{\|W_i - W_j\|} \quad \text{(as } W \text{ stays unchanged)} \\
&= \frac{|\langle W_i - W_j, \nu q\rangle + \nu(b_i - b_j)|}{\|W_i - W_j\|} \\
&= \nu \times \frac{|\langle W_i - W_j, q\rangle + (b_i - b_j)|}{\|W_i - W_j\|} = \nu d(q, \mathcal{P}_{ij}).
\end{aligned}
$$

So, the distance of $q$ to an arbitrary decision boundary is multiplied by $\nu$. As we consider an arbitrary input, by the definition of class margin, the class margin of all classes is multiplied by $\nu$.

**3. Now, let us consider the class dispersion.** Let consider two arbitrary points $q_1, q_2 \in \mathcal{F}$. The (euclidean) distance between these 2 points is $\|q_1 - q_2\|$. By Remark D.1, we know that $q_1$ and $q_2$ are turned into $\tilde{q}_1 := \nu q_1$ and $\tilde{q}_2 := \nu q_2$, respectively. Hence, the distance between these 2 points is now $\|\tilde{q}_1 - \tilde{q}_2\| = \nu\|q_1 - q_2\|$. So, the distance between $q_1$ and $q_2$ is multiplied by $\nu$. As we chose 2 arbitrary points, this holds for any 2 points. By the definition of class dispersion, it is obvious that the class dispersion of any class is multiplied by $\nu$ as well.

**4. Prove that $\mathcal{T}_\nu(\Theta) - \Theta = (\nu - 1)u$, where $u$ is a vector depending only on $\Theta$.** By definition of $\mathcal{T}_\nu$, we have that

$$
\begin{aligned}
\mathcal{T}_\nu(\Theta) - \Theta &= \begin{pmatrix} \Theta_I \\ \nu(W^-, b^-, b) \end{pmatrix} - \begin{pmatrix} \Theta_I \\ (W^-, b^-, b) \end{pmatrix} \\
&= \begin{pmatrix} \mathbf{0} \\ (\nu - 1)(W^-, b^-, b) \end{pmatrix} = (\nu - 1)\begin{pmatrix} \mathbf{0} \\ (W^-, b^-, b) \end{pmatrix} .
\end{aligned}
$$

So, we have that $\mathcal{T}_\nu(\Theta) - \Theta = (\nu - 1)u$, where $u = \begin{pmatrix} \mathbf{0} \\ (W^-, b^-, b) \end{pmatrix}$, depending only on $\Theta$.

## Appendix E  Proof of Theorem 6.1

In this section, we provide a complete proof for the bounds of the curvatures. For clarity, we introduce separately some technical lemmas that are useful for the main proof (Lemmas E.1, E.2, E.3 and E.4 in Subsection E.3).

### E.1  Proof of the upper-bound

**Proof E.1** *Using the property* $\mathrm{tr}(H^2) = \|H\|_F^2 = \sum_i \lambda_i^2$ *in Lemma E.1, it suffices to upper bound* $\mathrm{tr}(H^2)$. *Using the assumption on quadratic approximation of* $l(x + \varepsilon)$ *around* $x$, *we have:*

$$
l(x + \varepsilon) = l(x) + \nabla l(x)^\top \varepsilon + \frac{1}{2}\varepsilon^\top H \varepsilon .
$$

***Step 1.*** *Showing that* $\mathrm{tr}(H^2) \leq \frac{2}{\sigma^4}\mathbb{E}[|l(x + \varepsilon) - l(x)|^2]$.

*We analyze the squared loss change:*

$$
(l(x + \varepsilon) - l(x))^2 = \left( \nabla l(x)^\top \varepsilon + \frac{1}{2}\varepsilon^\top H \varepsilon \right)^2
$$

*Let's denote $A = \nabla l^\top \varepsilon, \quad B = \frac{1}{2}\varepsilon^\top H \varepsilon$. Take expectation:*

$$\mathbb{E}[(l(x+\varepsilon) - l(x))^2] = \mathbb{E}[A^2] + 2\mathbb{E}[AB] + \mathbb{E}[B^2] \ .$$

*For the cross-term, we have:*

$$\mathbb{E}[AB] = \frac{1}{2}\mathbb{E}[(\nabla l^\top \varepsilon)(\varepsilon^\top H \varepsilon)] = \frac{1}{2}\nabla l^\top \cdot \mathbb{E}[\varepsilon \varepsilon^\top H \varepsilon] \ .$$

*By Lemma E.2, we know that $\mathbb{E}[\varepsilon \varepsilon^\top H \varepsilon] = 0$. Hence,*

$$\mathbb{E}[AB] = 0 \ .$$

*For the quadratic term, by using Lemma E.3, we have*

$$\mathbb{E}[B^2] = \frac{1}{4}\mathbb{E}[(\varepsilon^\top H \varepsilon)^2] = \frac{1}{4} \cdot \sigma^4 \left(2\|H\|_F^2 + (\mathrm{tr}H)^2\right)$$

*So we get:*

$$\mathbb{E}[(l(x+\varepsilon) - l(x))^2] = \mathbb{E}[A^2] + \frac{1}{4}\sigma^4(2\|H\|_F^2 + (\mathrm{tr}H)^2) \geq \frac{\sigma^4}{2}\|H\|_F^2 \ . \tag{22}$$

*Therefore,*

$$\mathrm{tr}(H^2) = \|H\|_F^2 \leq \frac{2}{\sigma^4}\mathbb{E}[(l(x+\varepsilon) - l(x))^2] \ .$$

**Step 2.** *Partitioning the expectation of the squared loss change into 2 regions.*

*Let $\mathcal{A}$ be the event "$f_\theta(x+\varepsilon) \in \mathcal{C}(f(x), \delta)$". So, $\mathbb{P}(\mathcal{A}) = \eta$. We split the expectation into two regions $\mathcal{A}$ and $\overline{\mathcal{A}}$:*

$$\mathbb{E}\left[(l(x+\varepsilon) - l(x))^2\right] = \mathbb{E}\left[(l(x+\varepsilon) - l(x))^2 \mid \mathcal{A}\right] \cdot \eta + \mathbb{E}\left[(l(x+\varepsilon) - l(x))^2 \mid \overline{\mathcal{A}}\right] \cdot (1-\eta) \tag{23}$$

**Step 3.** *Showing that $|l(x+\varepsilon) - l(x)| \leq 2\max_j |\Delta z_j|$, where $\Delta z_j := z_j(x+\varepsilon) - z_j(x)$ is the logit shift.*

*Recall that the cross-entropy loss w.r.t. the true class $y$ is*

$$l(x) = -\log \frac{e^{z_y(x)}}{\sum_j e^{z_j(x)}} = -z_y(x) + \log \sum_j e^{z_j(x)}.$$

*Hence, let $p_j = \sigma(z)_j = e^{z_j}/\sum_i e^{z_i}$ (so $\sum_j p_j = 1$), we have*

$$l(x+\varepsilon) - l(x) = -\Delta z_y + \log \frac{\sum_j e^{z_j(x+\varepsilon)}}{\sum_j e^{z_j(x)}} = -\Delta z_y + \log \sum_j p_j e^{\Delta z_j} \ .$$

*Therefore,*

$$|l(x+\varepsilon) - l(x)| \leq |\Delta z_y| + \left|\log \sum_j p_j e^{\Delta z_j}\right|$$

As $\sum_j p_j = 1, p_j \geq 0$, the sum $\sum_j p_j e^{\Delta z_j}$ is a convex combination of exponentials. Hence, the log-sum-exp function satisfies

$$\min_j \Delta z_j \leq \log \sum_j p_j e^{\Delta z_j} \leq \max_j \Delta z_j .$$

Hence, $\left| \log \sum_j p_j e^{\Delta z_j} \right| \leq \max_j |\Delta z_j|$. Obviously, we also have $|\Delta z_y| \leq \max_j |\Delta z_j|$. So,

$$|l(x + \varepsilon) - l(x)| \leq 2 \max_j |\Delta z_j| .$$

**Step 4.** *Upper-bounding* $\max_j |\Delta z_j|$ *in the case where* $\mathcal{A}$ *happens.*

In this case, $f_\theta(x + \varepsilon) \in \mathcal{C}(f(x), \delta)$. It is easy to see that $\|f_\theta(x + \varepsilon) - f_\theta(x)\| \leq \delta$.

Recall that the logits for input $x$ is defined by

$$z(x) = W f_\theta(x) + b \in \mathbb{R}^C .$$

The logits are

$$z_j(x) = W_j^\top f_\theta(x) + b_j,$$

Let $v = f_\theta(x + \varepsilon) - f_\theta(x)$, so after perturbation,

$$z_j(x + \varepsilon) = W_j^\top f_\theta(x + \varepsilon) + b_j = z_j(x) + W_j^\top v.$$

Thus the logit shift $\Delta z_j$ is bounded by

$$|\Delta z_j| = |z_j(x + \varepsilon) - z_j(x)| \leq \|W_j\| \cdot \|v\| \leq \|W_j\| \, \delta.$$

Let

$$\|W\|_{2,\infty} := \max_j \|W_j\|.$$

Then for all $j$,

$$|\Delta z_j| \leq \|W\|_{2,\infty} \, \delta.$$

This means that $\max_j |\Delta z_j| \leq \|W\|_{2,\infty} \, \delta$.

**Step 5.** *Upper-bounding* $\max_j |\Delta z_j|$ *in the case where* $\mathcal{A}$ *does not happen. In this case, as we assume that the absolute value of logit is bounded by* $K_{\max}$, *it is easy to see that* $\max_j |\Delta z_j| \leq 2K_{\max}$.

**Step 6.** *Upper-bounding* $\mathbb{E}\left[(l(x + \varepsilon) - l(x))^2\right]$ *to upper-bound* $\mathrm{tr}(H^2)$.

From Eq. (23) and using the results of Steps 3, 4 and 5, we have that

$$\mathbb{E}\left[(l(x + \varepsilon) - l(x))^2\right] \leq 4\eta \|W\|_{2,\infty}^2 \, \delta^2 + 16(1 - \eta) \cdot K_{\max}^2 .$$

So

$$\mathrm{tr}(H^2) \leq \frac{8}{\sigma^4} \left( \eta \|W\|_{2,\infty}^2 \, \delta^2 + 4(1 - \eta) \cdot K_{\max}^2 \right) .$$

This completes the proof.

### E.2 Proof of the lower-bound

**Proof E.2 Step 1.** *Using the property* $\mathrm{tr}(H^2) = \|H\|_F^2 = \sum_i \lambda_i^2$ *and* $\mathrm{tr}(H) = \sum_i \lambda_i$ *in Lemma E.1, we first establish a relation between the terms relating to* $\mathrm{tr}(H)$ *and the expectation of the squared loss change.*

*From the result of Eq. (22) in previous part, we know that:*

$$\mathbb{E}[(l(x+\varepsilon) - l(x))^2] = \mathbb{E}[A^2] + \frac{1}{4}\sigma^4(2\|H\|_F^2 + (\mathrm{tr}H)^2) \ ,$$

*where $A = \nabla l^\top \varepsilon$. It is easy to see that $\mathbb{E}[A^2] = \sigma^2 \|\nabla l\|^2$.*

*So,*

$$2\|H\|_F^2 + (\mathrm{tr}H)^2 = \frac{4}{\sigma^4}\left(\mathbb{E}[(l(x+\varepsilon) - l(x))^2] - \sigma^2\|\nabla l\|^2\right) \ .$$

*We note that $\|H\|_F^2 = \sum_i \lambda_i^2$ and $|\mathrm{tr}(H)| = |\sum_i \lambda_i| \le \sum_i |\lambda_i|$, which leads to*

$$2\sum_i \lambda_i^2 + \left(\sum_i |\lambda_i|\right)^2 \ge \frac{4}{\sigma^4}\left(\mathbb{E}[(l(x+\varepsilon) - l(x))^2] - \sigma^2\|\nabla l\|^2\right) \ .$$

**Step 2.** *Partitioning the expectation of the squared loss change into 2 regions to find a lower bound.*

*Let $\mathcal{A}$ be the event "$f_\theta(x+\varepsilon) \in \mathcal{C}(f(x), \delta)$". So, $\mathbb{P}(\mathcal{A}) = \eta$. We split the expectation into two regions $\mathcal{A}$ and $\overline{\mathcal{A}}$:*

$$\mathbb{E}\left[(l(x+\varepsilon) - l(x))^2\right] = \mathbb{E}\left[(l(x+\varepsilon) - l(x))^2 \mid \mathcal{A}\right] \cdot \eta + \mathbb{E}\left[(l(x+\varepsilon) - l(x))^2 \mid \overline{\mathcal{A}}\right] \cdot (1-\eta)$$
$$\ge (1-\eta) \cdot \mathbb{E}\left[(l(x+\varepsilon) - l(x))^2 \mid \overline{\mathcal{A}}\right] \ .$$

*We notice that by Lemma E.4, we have:*

$$\mathbb{E}\left[(l(x+\varepsilon) - l(x))^2 \mid \overline{\mathcal{A}}\right] = Var\left(l(x+\varepsilon) \mid \overline{\mathcal{A}}\right) + \left(\mathbb{E}\left[l(x+\varepsilon) \mid \overline{\mathcal{A}}\right] - l(x)\right)^2$$
$$\ge \left(\mathbb{E}\left[l(x+\varepsilon) \mid \overline{\mathcal{A}}\right] - l(x)\right)^2$$
$$= (l_{out} - l(x))^2 \ \left(\mathbb{E}\left[l(x+\varepsilon) \mid \overline{\mathcal{A}}\right] = l_{out}\right) \ .$$

*Thus,*

$$\mathbb{E}\left[(l(x+\varepsilon) - l(x))^2\right] \ge (1-\eta) \cdot (l_{out} - l(x))^2 \ .$$

**Step 3. Conclusion.**

*Using the results of step 1 and step 2 leads us to the conclusion*

$$2\sum_i \lambda_i^2 + \left(\sum_i |\lambda_i|\right)^2 \ge \frac{4}{\sigma^4}\left((1-\eta) \cdot (l_{out} - l(x))^2 - \sigma^2\|\nabla l\|^2\right) \ .$$

*This completes the proof.*

### E.3 Some useful technical lemmas

We recall some classical lemmas (with complete proof) that are useful for the proof of Theorem 6.1.

**Lemma E.1** *Let $H \in \mathbb{R}^{n \times n}$ be a symmetric matrix with eigenvalues $\lambda_1, \lambda_2, \ldots, \lambda_n$. Then,*
$\mathrm{tr}(H) = \sum_{i=1}^n \lambda_i$ *and* $\mathrm{tr}(H^2) = \sum_{i=1}^n \lambda_i^2$.

**Proof E.3** *Since $H$ is symmetric, by the spectral theorem, there exists an orthogonal matrix $Q$ (i.e., $Q^T Q = I$) and a diagonal matrix $\Lambda = \mathrm{diag}(\lambda_1, \lambda_2, \ldots, \lambda_n)$ such that*

$$H = Q\Lambda Q^T.$$

*Then,*

$$\text{tr}(H) = \text{tr}(Q\Lambda Q^T) = \text{tr}(\Lambda Q^T Q) = \text{tr}(\Lambda I) = \text{tr}(\Lambda) = \sum_{i=1}^{n} \lambda_i$$

*Moreover,*

$$H^2 = (Q\Lambda Q^T)(Q\Lambda Q^T) = Q\Lambda(Q^T Q)\Lambda Q^T = Q\Lambda^2 Q^T,$$

*where $\Lambda^2 = \text{diag}(\lambda_1^2, \lambda_2^2, \ldots, \lambda_n^2)$.*

*Using the cyclic property of the trace, we have*

$$\text{tr}(H^2) = \text{tr}(Q\Lambda^2 Q^T) = \text{tr}(\Lambda^2 Q^T Q) = \text{tr}(\Lambda^2) = \sum_{i=1}^{n} \lambda_i^2.$$

*This completes the proof.*

**Lemma E.2** *Let $\varepsilon \sim \mathcal{N}(0, \sigma^2 I_n)$ and let $H \in \mathbb{R}^{n \times n}$ be a deterministic matrix. Then:*

$$\mathbb{E}[\varepsilon \varepsilon^\top H \varepsilon] = 0.$$

**Proof E.4** *We consider the $i^{th}$ component of the vector $\mathbb{E}[\varepsilon \varepsilon^\top H \varepsilon] \in \mathbb{R}^n$. It is given by:*

$$\left(\mathbb{E}[\varepsilon \varepsilon^\top H \varepsilon]\right)_i = \mathbb{E}\left[\sum_{j=1}^{n} \varepsilon_i \varepsilon_j (H\varepsilon)_j\right] = \sum_{j=1}^{n} \mathbb{E}[\varepsilon_i \varepsilon_j (H\varepsilon)_j].$$

*Note that*

$$(H\varepsilon)_j = \sum_{k=1}^{n} H_{jk} \varepsilon_k,$$

*so we can write:*

$$\left(\mathbb{E}[\varepsilon \varepsilon^\top H \varepsilon]\right)_i = \sum_{j=1}^{n} \sum_{k=1}^{n} H_{jk} \mathbb{E}[\varepsilon_i \varepsilon_j \varepsilon_k].$$

*Now, since $\varepsilon \sim \mathcal{N}(0, \sigma^2 I_n)$, the random variables $\varepsilon_i$ are jointly Gaussian with zero mean. Therefore, any odd-order moment (i.e., the product of an odd number of Gaussian variables) has expectation zero:*

$$\mathbb{E}[\varepsilon_i \varepsilon_j \varepsilon_k] = 0 \quad \text{for all } i, j, k.$$

*Hence, every term in the above double sum vanishes, and we conclude:*

$$\left(\mathbb{E}[\varepsilon \varepsilon^\top H \varepsilon]\right)_i = 0 \quad \text{for all } i.$$

*Therefore,*

$$\mathbb{E}[\varepsilon \varepsilon^\top H \varepsilon] = 0.$$

**Lemma E.3** *Let $\varepsilon \sim \mathcal{N}(0, \sigma^2 I_n)$ and $H \in \mathbb{R}^{n \times n}$ be a symmetric matrix. Then,*

$$\mathbb{E}[(\varepsilon^\top H \varepsilon)^2] = 2\sigma^4 \|H\|_F^2 + \sigma^4 (\text{tr}H)^2 .$$

**Proof E.5** *Let $\varepsilon \sim \mathcal{N}(0, \sigma^2 I_n)$ and $H \in \mathbb{R}^{n \times n}$ be a symmetric matrix. We want to compute*

$$\mathbb{E}[(\varepsilon^\top H \varepsilon)^2].$$

*Since $H$ is symmetric, it can be diagonalized as*

$$H = Q\Lambda Q^\top,$$

where $Q$ is an orthogonal matrix and $\Lambda = \mathrm{diag}(\lambda_1, \ldots, \lambda_n)$ contains the eigenvalues of $H$.

Define $\tilde{\varepsilon} := Q^\top \varepsilon$. Then $\tilde{\varepsilon} \sim \mathcal{N}(0, \sigma^2 I)$ and we have

$$\varepsilon^\top H \varepsilon = \varepsilon^\top Q \Lambda Q^\top \varepsilon = \tilde{\varepsilon}^\top \Lambda \tilde{\varepsilon} = \sum_{i=1}^{n} \lambda_i \tilde{\varepsilon}_i^2.$$

Define $X := \sum_{i=1}^{n} \lambda_i \tilde{\varepsilon}_i^2$. Then:

$$\mathbb{E}[X^2] = \mathbb{E}\left[ \left( \sum_{i=1}^{n} \lambda_i \tilde{\varepsilon}_i^2 \right)^2 \right] = \sum_{i,j} \lambda_i \lambda_j \mathbb{E}[\tilde{\varepsilon}_i^2 \tilde{\varepsilon}_j^2].$$

Now recall:

$$\mathbb{E}[\tilde{\varepsilon}_i^4] = 3\sigma^4, \quad \text{for all } i,$$
$$\mathbb{E}[\tilde{\varepsilon}_i^2 \tilde{\varepsilon}_j^2] = \mathbb{E}[\tilde{\varepsilon}_i^2]\mathbb{E}[\tilde{\varepsilon}_j^2] = \sigma^4, \quad \text{for } i \neq j.$$

Therefore,

$$\mathbb{E}[X^2] = \sum_{i=1}^{n} \lambda_i^2 \cdot 3\sigma^4 + \sum_{i \neq j} \lambda_i \lambda_j \cdot \sigma^4$$
$$= 3\sigma^4 \sum_{i=1}^{n} \lambda_i^2 + \sigma^4 \left( \sum_{i,j} \lambda_i \lambda_j - \sum_{i=1}^{n} \lambda_i^2 \right)$$
$$= 3\sigma^4 \|H\|_F^2 + \sigma^4 \left( (\mathrm{tr}H)^2 - \|H\|_F^2 \right)$$
$$= 2\sigma^4 \|H\|_F^2 + \sigma^4 (\mathrm{tr}H)^2.$$

So, $\mathbb{E}[(\varepsilon^\top H \varepsilon)^2] = 2\sigma^4 \|H\|_F^2 + \sigma^4 (\mathrm{tr}H)^2$ .

**Lemma E.4** *Let $X \in \mathbb{R}$ be a real-valued random variable, and let $A$ be an event (or more generally, a sub-$\sigma$-algebra). Let $\rho \in \mathbb{R}$ be a fixed constant. Then:*

$$\mathbb{E}\left[ (X - \rho)^2 \mid A \right] = \mathrm{Var}(X \mid A) + (\mathbb{E}[X \mid A] - \rho)^2.$$

### Proof E.6

Let $\mu := \mathbb{E}[X \mid A]$. Then we can write:

$$X - \rho = (X - \mu) + (\mu - \rho),$$

and hence:

$$(X - \rho)^2 = (X - \mu)^2 + 2(X - \mu)(\mu - \rho) + (\mu - \rho)^2.$$

Taking conditional expectation with respect to $A$, and using linearity of expectation:

$$\mathbb{E}[(X - \rho)^2 \mid A] = \mathbb{E}[(X - \mu)^2 \mid A] + 2(\mu - \rho)\mathbb{E}[X - \mu \mid A] + (\mu - \rho)^2.$$

Since $\mu = \mathbb{E}[X \mid A]$, we have $\mathbb{E}[X - \mu \mid A] = 0$, and thus:

$$\mathbb{E}[(X - \rho)^2 \mid A] = \mathbb{E}[(X - \mu)^2 \mid A] + (\mu - \rho)^2 = \mathrm{Var}(X \mid A) + (\mathbb{E}[X \mid A] - \rho)^2.$$

## Appendix F  Proofs of generalization errors

In this section, we provide the complete proofs concerning the generalization errors discussed in Section 6.2. Our proofs are inspired by the methodology developed in Mohri (2018). To begin with, we introduce some notations and results given in Mohri (2018).

**Definition F.1 (Empirical Rademacher complexity, p. 30 in Mohri (2018))** *Let $H$ be a family of functions mapping from $\mathcal{Z}$ to $[a, b]$ and $S = (z_1, ..., z_N)$ a fixed sample of size $N$ with elements in $\mathcal{Z}$. Then, the empirical Rademacher complexity of $H$ with respect to the sample $S$ is defined as:*

$$\widehat{\mathcal{R}}_S(H) = \mathbb{E}_\sigma \left[ \sup_{h \in H} \frac{1}{N} \sum_{i=1}^N \sigma_i h(z_i) \right], \tag{24}$$

*where $\sigma = (\sigma_1, ..., \sigma_N)$, with $\sigma_i$'s independent uniform random variables taking values in $\{-1, +1\}$. The random variables $\sigma_i$ are called Rademacher variables.*

**Theorem F.1 (Theorem 3.3, p. 31 in Mohri (2018))** *Let $\mathcal{G}$ be a family of functions mapping from $\mathbb{Z}$ to $[0, 1]$. Then, for any $\delta > 0$, with probability at least $1 - \delta$ over the draw of an i.i.d. sample $S$ of size $N$, the following holds for all $g \in \mathcal{G}$:*

$$\mathbb{E}[g(Z)] \leq \frac{1}{N} \sum_{i=1}^N g(z_i) + 2\widehat{\mathcal{R}}_S(\mathcal{G}) + 3\sqrt{\frac{\log \frac{2}{\delta}}{2N}} \ .$$

**Lemma F.1 (Talagrand's lemma, p.93 in Mohri (2018))** *Let $\Phi : \mathbb{R} \mapsto \mathbb{R}$ be an $l$-Lipschitz $(l > 0)$. Then, for any hypothesis set $H$ of real-valued functions, the following inequality holds:*

$$\widehat{\mathcal{R}}_S(\Phi \circ H) \leq l\widehat{\mathcal{R}}_S(H) \ .$$

.

Using Theorem F.1 and Lemma F.1, we can derive following theorem:

**Theorem F.2** *Let $H$ be a set of real-valued functions and let $P_X$ be the distribution over the input space $\mathcal{X}$. Fix $\rho > 0$, then, for any $\delta > 0$, with probability at least $1 - \delta$ over a sample $S$ of size $N$ drawn according to $P_X$, the following holds for all $h \in H$:*

$$R(h) \leq \widehat{R}_{S,\rho}(h) + \frac{2}{\rho}\widehat{\mathcal{R}}_S(H) + 3\sqrt{\frac{\log \frac{2}{\delta}}{2N}} \ , \tag{25}$$

*where $R(h) = \mathbb{E}[1_{\{h(X) < 0\}}]$ (generalization error), $\widehat{R}_{S,\rho}(h) = \frac{1}{N} \sum_{i=1}^N \Phi_\rho(h(x_i))$ is the empirical margin loss on $S$ of size $N$ and $\widehat{\mathcal{R}}_S(H)$ is the empirical Rademacher complexity of $H$ with respect to the sample $S$.*

**Proof F.1** *Let $\mathcal{G} = \{\Phi_\rho \circ h : h \in H\}$. By theorem F.1, for all $g \in \mathcal{G}$,*

$$\mathbb{E}[g(X)] \leq \frac{1}{N} \sum_{i=1}^N g(x_i) + 2\widehat{\mathcal{R}}_S(\mathcal{G}) + 3\sqrt{\frac{\log \frac{2}{\delta}}{2N}} \ ,$$

*and consequently, for all $h \in H$,*

$$\mathbb{E}[\Phi_\rho(h(X))] \leq \frac{1}{N} \sum_{i=1}^N \Phi_\rho(h(x_i)) + 2\widehat{\mathcal{R}}_S(\Phi_\rho \circ H) + 3\sqrt{\frac{\log \frac{2}{\delta}}{2N}} \ .$$

*Since $1_{\{u \leq 0\}} \leq \Phi_\rho(u)$, we have $R(h) = \mathbb{E}[1_{\{h(X) < 0\}}] \leq \mathbb{E}[\Phi_\rho(h(X))]$, so*

$$R(h) \leq \widehat{R}_{S,\rho}(h) + 2\widehat{\mathcal{R}}_S(\Phi_\rho \circ H) + 3\sqrt{\frac{\log \frac{2}{\delta}}{2N}} \ .$$

*On the other hand, since $\Phi_\rho$ is $1/\rho$-Lipschitz, by lemma F.1, we have $\widehat{\mathcal{R}}_S(\Phi_\rho \circ H) \leq \frac{1}{\rho}\widehat{\mathcal{R}}_S(H)$. So, $R(h) \leq \widehat{R}_{S,\rho}(h) + \frac{2}{\rho}\widehat{\mathcal{R}}_S(H) + 3\sqrt{\frac{\log\frac{2}{\delta}}{2N}}$ .*

With all these notions and theorems, we are well-equipped to prove Theorem 6.2.

### F.1 Proof of Theorem 6.2

By using Theorem F.2, to prove Theorem 6.2, it suffices to prove the following result:

**Theorem F.3** *The empirical Rademacher complexity of $H$ can be bounded as follows:*

$$\widehat{\mathcal{R}}_S(H) \leq \Lambda^2 + 2R\Lambda + \frac{R^2}{\sqrt{N}} \ . \tag{26}$$

**Proof F.2** *Let $\mathcal{G}_2 = \{f : \|\sup_{x \in \mathcal{X}} f(x)\| \leq \Lambda\}$. Let $\sigma = (\sigma_1, ..., \sigma_N)$, with $\sigma_i$'s independent uniform random variables taking values in $\{-1, +1\}$. By definition of the empirical Rademacher complexity, we have:*

$$
\begin{aligned}
&\widehat{\mathcal{R}}_S(H) \\
&= \mathbb{E}_\sigma \left[ \sup_{\|m\| \leq R, f \in \mathcal{G}_2} \frac{1}{N} \sum_{i=1}^N \sigma_i(r^2 - \|f(x_i) - m\|^2) \right] \\
&= \frac{1}{N}\mathbb{E}_\sigma \left[ \sum_{i=1}^N \sigma_i r^2 + \sup_{\|m\| \leq R, f \in \mathcal{G}_2} \sum_{i=1}^N -\sigma_i\|f(x_i) - m\|^2 \right] \\
&= \frac{1}{N}\mathbb{E}_\sigma \left[ \sup_{\|m\| \leq R, f \in \mathcal{G}_2} \sum_{i=1}^N -\sigma_i\|f(x_i) - m\|^2 \right] \quad \left( as\ \mathbb{E}_\sigma \left[ \sum_{i=1}^N \sigma_i r^2 \right] = r^2 \sum_{i=1}^N \mathbb{E}_\sigma[\sigma_i] = 0 \right) \\
&= \frac{1}{N}\mathbb{E}_\sigma \left[ \sup_{\|m\| \leq R, f \in \mathcal{G}_2} \sum_{i=1}^N (-\sigma_i\|f(x_i)\|^2 - \sigma_i\|m\|^2 + 2\sigma_i\langle f(x_i), m \rangle) \right] \\
&\leq \frac{1}{N}\mathbb{E}_\sigma \left[ \sup_{f \in \mathcal{G}_2} \sum_{i=1}^N -\sigma_i\|f(x_i)\|^2 + \sup_{\|m\| \leq R} \sum_{i=1}^N -\sigma_i\|m\|^2 + \sup_{\|m\| \leq R, f \in \mathcal{G}_2} \sum_{i=1}^N 2\sigma_i\langle f(x_i), m \rangle \right] \ .
\end{aligned}
$$

*Considering each term inside the expectation operation, we get,*

$$
\begin{aligned}
\mathbb{E}_\sigma \left[ \sup_{f \in \mathcal{G}_2} \sum_{i=1}^N -\sigma_i\|f(x_i)\|^2 \right] &\leq \mathbb{E}_\sigma \left[ \sup_{f \in \mathcal{G}_2} |\sum_{i=1}^N -\sigma_i\|f(x_i)\|^2| \right] \leq \mathbb{E}_\sigma \left[ \sup_{f \in \mathcal{G}_2} \sum_{i=1}^N |\sigma_i\|f(x_i)\|^2| \right] \\
&= \mathbb{E}_\sigma \left[ \sup_{f \in \mathcal{G}_2} \sum_{i=1}^N \|f(x_i)\|^2 \right] \leq \mathbb{E}_\sigma \left[ \sum_{i=1}^N \Lambda^2 \right] = N\Lambda^2 \ .
\end{aligned}
$$

*Consider now the second term. We have,*

$$
\mathbb{E}_\sigma \left[ \sup_{\|m\|\leq R} \sum_{i=1}^N -\sigma_i\|m\|^2 \right] \leq \mathbb{E}_\sigma \left[ \sup_{\|m\|\leq R} \left| \sum_{i=1}^N -\sigma_i\|m\|^2 \right| \right] = \mathbb{E}_\sigma \left[ \sup_{\|m\|\leq R} \|m\|^2 \left| \sum_{i=1}^N \sigma_i \right| \right]
$$

$$
\leq R^2 \mathbb{E}_\sigma \left[ \left| \sum_{i=1}^N \sigma_i \right| \right] \leq R^2 \left( \mathbb{E}_\sigma \left[ \left| \sum_{i=1}^N \sigma_i \right|^2 \right] \right)^{\frac{1}{2}}
$$

$$
= R^2 \left( \mathbb{E}_\sigma \left[ \sum_{i,j=1}^N \sigma_i\sigma_j \right] \right)^{\frac{1}{2}}
$$

$$
= R^2 \left( \mathbb{E}_\sigma \left[ \sum_{i=1}^N \sigma_i^2 \right] \right)^{\frac{1}{2}} \quad (\text{ as } \mathbb{E}_\sigma[\sigma_i\sigma_j] = 0 \text{ if } i \neq j \text{ and } 1 \text{ otherwise})
$$

$$
= R^2\sqrt{N} \ .
$$

*Consider now the final term inside the expectation operation.*

$$
\mathbb{E}_\sigma \left[ \sup_{\|m\|\leq R, f\in\mathcal{G}_2} \sum_{i=1}^N 2\sigma_i\langle f(x_i), m\rangle \right] \leq \mathbb{E}_\sigma \left[ \sup_{\|m\|\leq R, f\in\mathcal{G}_2} \left| \sum_{i=1}^N 2\sigma_i\langle f(x_i), m\rangle \right| \right]
$$

$$
\leq \mathbb{E}_\sigma \left[ \sup_{\|m\|\leq R, f\in\mathcal{G}_2} \left| \sum_{i=1}^N 2\sigma_i\|f(x_i)\| \cdot \|m\| \right| \right]
$$

$$
\leq \mathbb{E}_\sigma \left[ 2R \sup_{f\in\mathcal{G}_2} \sum_{i=1}^N |\sigma_i| \cdot \|f(x_i)\| \right]
$$

$$
= \mathbb{E}_\sigma \left[ 2R \sup_{f\in\mathcal{G}_2} \sum_{i=1}^N \|f(x_i)\| \right] \leq \mathbb{E}_\sigma \left[ 2R \sum_{i=1}^N \Lambda \right] = 2NR\Lambda \ .
$$

*Hence, $\widehat{\mathcal{R}}_S(H) \leq \frac{1}{N}(\Lambda^2 N + R^2\sqrt{N} + 2NR\Lambda) = \Lambda^2 + 2R\Lambda + \frac{R^2}{\sqrt{N}}$ .*

## Appendix G   Some Definitions and Proof of Theorem 6.3

We first recall some definitions and lemmas needed in the proof.

**Definition G.1 (Kullback–Leibler divergence)** *Let $\mathcal{D}_1$ and $\mathcal{D}_2$ be probability measures on a measurable space $\Omega$. The Kullback–Leibler divergence (KL divergence) between $\mathcal{D}_1$ and $\mathcal{D}_2$ is defined as $D_{KL}(\mathcal{D}_1||\mathcal{D}_2) = \int_{\omega\in\Omega} \log\left(\frac{d\mathcal{D}_1}{d\mathcal{D}_2}(\omega)\right) \mathcal{D}_1(d\omega)$ , and $+\infty$ if the Radon-Nikodym derivative $\frac{d\mathcal{D}_1}{d\mathcal{D}_2}$ does not exist.*

We note that $D_{KL}$ is not symmetric. To overcome such drawbacks, we can use the Jensen-Shannon divergence with the following definition.

**Definition G.2 (Jensen–Shannon divergence (JSD))** *Let $\mathcal{D}_1$ and $\mathcal{D}_2$ be probability measures on a measurable space $\Omega$. The Jensen–Shannon divergence (JSD) between $\mathcal{D}_1$ and $\mathcal{D}_2$ is defined as $D_{JS}(\mathcal{D}_1||\mathcal{D}_2) = \frac{1}{2}D_{KL}(\mathcal{D}_1||\bar{\mathcal{D}}) + \frac{1}{2}D_{KL}(\mathcal{D}_2||\bar{\mathcal{D}})$ , where $\bar{\mathcal{D}} = \frac{1}{2}(\mathcal{D}_1 + \mathcal{D}_2)$ is a mixture distribution of $\mathcal{D}_1$ and $\mathcal{D}_2$.*

**Definition G.3 (Total variation distance)** *Let $\mathcal{D}_1$ and $\mathcal{D}_2$ be probability measures on a measurable space $\Omega$. The total variation (TV) distance between $\mathcal{D}_1$ and $\mathcal{D}_2$ is defined as*

$$
d_{TV}(\mathcal{D}_1, \mathcal{D}_2) = \sup_{S\subseteq\Omega} |\mathcal{D}_1(S) - \mathcal{D}_1(S)| \ .
$$

**Lemma G.1 (Pinsker's inequality)** *For every $\mathcal{D}_1$, $\mathcal{D}_2$ on $\Omega$,*

$$d_{TV}(\mathcal{D}_1, \mathcal{D}_2) \leq \sqrt{\frac{1}{2}D_{KL}(\mathcal{D}_1 || \mathcal{D}_2)} \ .$$

**Lemma G.2 (Relation between JSD and mutual information)** *Let $B$ be the equiprobable binary r.v. taking value in $\{0, 1\}$. Given two distributions $\mathcal{D}_1$ and $\mathcal{D}_2$, let $X_1 \sim \mathcal{D}_1$ let $X_2 \sim \mathcal{D}_2$. Assume that $X_1$ and $X_2$ are independent of $B$, let $X = BX_1 + (1 - B)X_2$. Then, we have*

$$D_{JS}(\mathcal{D}_1 || \mathcal{D}_2) = I(X, B) \ , \tag{27}$$

*where $I(X, B)$ is the mutual information of $X$ and $B$ (see Cover & Thomas (2006)).*

**Remark.** We see that we choose the value of $X$ according to $\mathcal{D}_1$ if $B = 0$ and according to $\mathcal{D}_2$ if $B = 1$. That is, $\mathcal{D}(X|B = 0) = \mathcal{D}_1$ and $\mathcal{D}(X|B = 1) = \mathcal{D}_2$. Hence, $B$ can be considered as "flipping variable". Moreover, notice that in Eq. (27), the entropy (used in the mutual information) is with respect to a reference measure and that implicitly $\mathcal{D}_1$ and $\mathcal{D}_2$ are absolutely continuous with respect to this reference measure.

Now, we are ready for the proof.

**Proof G.1** *We consider any class pair $(y \neq y') \in \mathcal{Y}^2$. Let us first consider the conditional distributions on $\mathcal{F}$. For brevity, we write $Q$ instead of $Q^L$. We have that*

$$\begin{aligned} d_{TV}\left(\mathcal{D}_{Q|y}, \frac{\mathcal{D}_{Q|y} + \mathcal{D}_{Q|y'}}{2}\right) &= \sup_{S \subseteq \mathcal{F}} \left| \mathcal{D}_{Q|y}(S) - \frac{\mathcal{D}_{Q|y}(S) + \mathcal{D}_{Q|y'}(S)}{2} \right| \\ &= \sup_{S \subseteq \mathcal{F}} \left| \frac{\mathcal{D}_{Q|y}(S) - \mathcal{D}_{Q|y'}(S)}{2} \right| . \end{aligned}$$

*We consider $S = \mathcal{C}(m_y, \delta_v)$, the hyper-sphere of radius $\delta_v$ centered at $m_y$. Recall that we have assumption $\mathcal{D}_{Q|y}(\mathcal{C}(m_y, \delta_v)) \geq \tau, \ \forall y \in \mathcal{Y} \ (1/2 \leq \tau \leq 1)$. Hence, we have $\mathcal{D}_{Q|y}(S) \geq \tau$ .*

*Moreover, as the hyper-spheres of each class are disjoint, $S = \mathcal{C}(m_y, \delta_v) \subseteq \mathcal{F} \setminus \mathcal{C}(m_{y'}, \delta_v)$. So, $\mathcal{D}_{Q|y'}(S) \leq \mathcal{D}_{Q|y'}(\mathcal{F} \setminus \mathcal{C}(m_{y'}, \delta_v)) = 1 - \mathcal{D}_{Q|y'}(\mathcal{C}(m_{y'}, \delta_v)) \leq 1 - \tau$ .*

*Therefore, we have that $\mathcal{D}_y(S) - \mathcal{D}_{y'}(S) \geq \tau - (1 - \tau) = 2\tau - 1$. Note that this term is positive as $\tau \geq 1/2$. Hence, $|\mathcal{D}_y(S) - \mathcal{D}_{y'}(S)| = \mathcal{D}_y(S) - \mathcal{D}_{y'}(S) \geq 2\tau - 1$.*

*So, $d_{TV}\left(\mathcal{D}_{Q|y}, \frac{\mathcal{D}_{Q|y} + \mathcal{D}_{Q|y'}}{2}\right) = \sup_{S \subseteq \mathcal{F}} |\mathcal{D}_{Q|y}(S) - \mathcal{D}_{Q|y'}(S)|/2 \geq (2\tau - 1)/2$ .*

*Using Lemma G.1, we have that*

$$\frac{1}{2}D_{KL}\left(\mathcal{D}_{Q|y} \ || \ \frac{\mathcal{D}_{Q|y} + \mathcal{D}_{Q|y'}}{2}\right) \geq d_{TV}\left(\mathcal{D}_{Q|y}, \frac{\mathcal{D}_{Q|y} + \mathcal{D}_{Q|y'}}{2}\right) \geq \frac{(2\tau - 1)^2}{4} \ .$$

*As this hold for any $(y \neq y') \in \mathcal{Y}$, we also have that*

$$\frac{1}{2}D_{KL}\left(\mathcal{D}_{Q|y'} \ || \ \frac{\mathcal{D}_{Q|y} + \mathcal{D}_{Q|y'}}{2}\right) \geq \frac{(2\tau - 1)^2}{4} \ .$$

*Therefore, by summing these 2 terms, using the definition of JSD, we have*

$$D_{JS}\left(\mathcal{D}_{Q|y} \ || \ \mathcal{D}_{Q|y'}\right) \geq \frac{(2\tau - 1)^2}{2} \ . \tag{28}$$

*Let $B$ be the equiprobable binary r.v. taking value in $\{0, 1\}$. Consider now the conditional distributions $\mathcal{D}_{Q^l|y}$ and $\mathcal{D}_{Q^l|y'}$ in the feature space $\mathcal{F}^l$ of a layer $l$ before the penultimate layer. Let $Q_B^l$ be the random variable*

*associated to a mixture distribution of $\mathcal{D}_{Q^l|y}$ and $\mathcal{D}_{Q^l|y'}$ with $B$ as flipping variable in the way defined in Lemma G.2. So, we have*

$$D_{JS}\left(\mathcal{D}_{Q^l|y} \,||\, \mathcal{D}_{Q^l|y'}\right) = I(B, Q_B^l) . \tag{29}$$

*Let $f_l$ denote the mapping representing the set of layers between layer $l$ and the penultimate layer. So, we have that the distributions $\mathcal{D}_{Q|y}$ and $\mathcal{D}_{Q|y'}$ are induced by $f_l$ from $\mathcal{D}_{Q^l|y}$ and $\mathcal{D}_{Q^l|y'}$, respectively. Let $Q_B = f_l(Q_B^l)$. So we have the Markovian relationship: $B \longrightarrow Q_B^l \longrightarrow Q_B$. The data processing inequality Cover & Thomas (2006) follows that*

$$I(B, Q_B^l) \geq I(B, Q_B) . \tag{30}$$

*Moreover, note that the distribution $\mathcal{D}_{Q_B}$ is induced by $f_l$ from $\mathcal{D}_{Q_B^l}$. Given $B = 0$, $\mathcal{D}(Q_B^l|B = 0) = \mathcal{D}_{Q^l|y}$. So, the distribution $\mathcal{D}(Q_B|B = 0)$ is induced from $\mathcal{D}_{Q^l|y}$ by $f_l$. That is, $\mathcal{D}(Q_B|B = 0) = \mathcal{D}_{Q|y}$. Similarly, we also have $\mathcal{D}(Q_B|B = 1) = \mathcal{D}_{Q|y'}$. Hence, by Lemma G.2, we have*

$$D_{JS}\left(\mathcal{D}_{Q|y} \,||\, \mathcal{D}_{Q|y'}\right) = I(B, Q_B) . \tag{31}$$

*From (29), (30) and (31), we have $D_{JS}\left(\mathcal{D}_{Q^l|y} \,||\, \mathcal{D}_{Q^l|y'}\right) \geq D_{JS}\left(\mathcal{D}_{Q|y} \,||\, \mathcal{D}_{Q|y'}\right)$. Combining with (28), we have that*

$$D_{JS}\left(\mathcal{D}_{Q^l|y} \,||\, \mathcal{D}_{Q^l|y'}\right) \geq \frac{(2\tau - 1)^2}{2} .$$

*This completes the proof.*

## Appendix H  Experiment details

In this section, we provide the details for conducting our experiments. All of our experiments are performed using *Pytorch*. When the models are trained, we fix the models and test with different noise levels and perturbations. We run 10 independent perturbations to report the results.

### H.1  Training ResNet18

For ResNet18, the features of the final convolutional layer are subjected to *Global Average Pooloing (GAP)* over the spatial dimensions to obtain a vector of dimension 512. Then this vector is passed through a *ReLU* non-lineariy, followed by a fully connected layer (output dim=128). The output space of this layer is the feature space $\mathcal{F}$ in our framework. Finally, this is passed through the softmax layer (with 10 output classes), which is composed of an affine transformation and softmax function.

We intentionally aimed to keep a very minimal setup. Thus, we use the same training scheme for all the experiments with ResNet18, with softmax loss and our loss, on both CIFAR10 and SVHN. We use SGD optimizer with nesterov momentum, with initial learning rate of $10^{-3}$ (decayed by factor of 5 at epochs 60,120 and 160), momentum 0.9, weight decay $5 \cdot 10^{-4}$. Model is trained for 200 epochs, with mini-batch size equal to 64.

For the training sets of CIFAR10 and SVHN, we apply data augmentation techniques including Random Cropping (padding=4) and Random Horizontal Flipping before data normalization. This is referred to as normal training in our main text. For the other methods using Gaussian noise, we simply add Gaussian noise on top of these data augmentations.

For our method, we set $\delta_v = 0.5$ and $\delta_d = 5.0$ in our constraints. The total loss is $\mathcal{L}_{\text{total}} = \mathcal{L}_{\text{ours}} + \mathcal{L}_{\mathcal{S}}$, where $\mathcal{L}_{\mathcal{S}}$ is the standard softmax loss (see Eq. (3)).

### H.2  Training MobileNetV3 on our road image dataset

We use models pretrained on *ImageNet* as initialization as our dataset is quite small to train the model from scratch.

During training, we apply data augmentation techniques including: ColorJitter(brightness=0.3, contrast=0.3,saturation=0.3), RandomRotation(10), Random Perspective (with pytorch default parameters)and Random Horizontal Flipping, before the data normalization. This is referred to as normal training in our main text. For the other methods using Gaussian noise, we simply add Gaussian noise on top of these data augmentations.

We use Adam optimizer with learning rate $10^{-4}$ (decayed by factor of 10 at epochs 50). We train models for 70 epochs, with and keep the best models based on the validation accuracy.

For our method, we set $\delta_v = 0.5$ and $\delta_d = 3.0$ in our constraints. The total loss is $\mathcal{L}_{\text{total}} = \mathcal{L}_{\text{ours}} + \mathcal{L}_{\mathcal{S}}$, where $\mathcal{L}_{\mathcal{S}}$ is the standard softmax loss (see Eq. (3)).

