# OpenReview forum: "Training More Robust Classification Model via Discriminative Loss and Gaussian Noise Injection"
_TMLR — Accepted by TMLR_

### Review · Reviewer_hc69 · 2025-10-30

**Summary Of Contributions:**

With the intent of improving feature discriminativeness and robustness against input noise, the authors of "Training More Robust Classification Model via Discriminative Loss and Gaussian Noise Injection" discuss a loss function that combines several existing loss formulations and apply it to the penultimate layer of a neural network. Their loss function specifically addresses the issues of **class dispersion, class margins and model robustness** against input feature noise. Fundamentally, it encourages the network to map embeddings of a class into the vicinity of its corresponding class centroid in the embedding space. Assuming optimal conditions (e.g., a loss being zero), the authors analyze theoretical properties of the neural network induced by their loss functions. Thereby, they derive an upper bound for class dispersions in the embedding space and a lower bound for class margins, both expressable in terms of loss hyperparameters. The authors further argue that optimizing solely for either minimal class dispersions or maximal class margins is a suboptimal, and they formalize a link between input loss curvature and  feature- as well as loss stability under Gaussian input noise. The theoretical part of this work concludes with a discussion about generalization properties and the impact of the loss on feature representations prior to the penultimate layer. The experimental part empirically validates the theoretical findings: 3 datasets (CIFAR-10, Street View House Numbers (SVHN) and a self-created dataset referred to as "Road Condition Image Dataset") are used to demonstrate performance improvements towards 4 baseline methods, to experimentally show the relation of model performance to input loss curvature and finally to visualize the effects on intermediate feature representations.

- **Strengths:**
	- *S1:* The proposed loss function allows for in-depth theoretical research on lower- and upper bounds for embedding space projections in terms of loss hyperparameters, and for establishing theoretical connections between model performance and input loss curvature.
	- *S2:* The structure of the theoretical part of this work allows for targeted experiment designs that precisely analyze theoretical claims.
	- *S3:* The experiment results validate claims anticipated by the theory.
	- *S4:* Experiment code has been provided.
- **Weaknesses**:
	- *W1:* The overall readability of the paper requires improvement. Detailed points are provided in the additional comments.
	- *W2:* The following references do not exist:
		- Rong Jin, Chiyuan Zhang, Cho-Jui Zhang, Qiang Tang, and Le Song. Input hessian regularization for adversarial robustness. In NeurIPS Workshop on Robust Machine Learning, 2019.
		- Seyed-Mohsen Moosavi-Dezfooli, Alhussein Fawzi, and Pascal Frossard. A second-order approach to adversarial attacks and defenses. arXiv preprint arXiv:1902.02864, 2019a.
	- *W3:* Proposition 5.3 claims more than the paper proves:
		- Proposition 5.3 claims to be applicable to fully-connected layers or to arbitrary convolution layers.
		- The proof only considers fully connected layers or image convolutions. In case the term "convolution" should be kept, it needs to be explicitly specified what sort of convolutions are considered. E.g., the definition of an image convolution is different from that of a graph-convolution.

**Additional Comments:**

**Aspects in the paper that require clarification:**
1. The work makes three core claims: the proposed loss function shall (a) enforce intra-class compactness (b) increase class margins and (c) increase model robustness. While the manuscript investigates these claims in great detail, it includes a few digressions. E.g., Sections 5.2 or 6.2 could be moved to the appendix.
2. About Equation 13:
	- This equation describes the overall loss proposed by the authors. It should be pointed out that this loss requires at least two forward passes, given that $L_{\text{clean}}$ requires clean- and $L_{\text{noisy}}$ requires noisy samples.
3. About Proposition 5.3:
	- $T_\alpha$ is **defined** as a linear shift and thus assumed to be present. Conversely, the change of network parameters is not necessarily something that has been observed as a training dynamic. This should be emphasized in Section 5.2.
	- It remains questionable whether the addressed methods in the second paragraph of Section 5.2 really cause a linear shift $T_\alpha$ of the model parameters as assumed by the proof.
4. About Theorem 6.3:
	- The authors write: "That is, in the intermediate layers, features of different classes become more separated from each other.". This phrasing suggests that Theorem 6.3 implies an increasing inter-class distance across layers as experimentally visualized in Section 7.5. However, it merely describes the existence of a lower bound for the JSD between two class distributions at layer $l$. In particular, it does not guarantee increasing margins between class centroids across layers.
5. About Section 5.2:
	- The analogy to Huygens Inertia Theorem seems to be far fetched. The authors should either choose an analogy more common to the ML-domain or introduce the Inertia Theorem sufficiently well.
6. Equations 8 and 9:
	- These equations introduce the hyperparameter $\delta_v$ and $\delta_d$, which are repeatedly used throughout the manuscript. Their naming should more clearly convey their respective meanings.
7. Below Equation 10 it reads: "However, there are no guarantee that the decision boundaries lead to closed cells.".
	- The manuscript should clarify what is understood as a "closed cell".
8. Below Proposition 4.1 it reads: "But at the same time, the centroids are kept consistent. That is, the parameters of the neural network evolve along training with sufficiently large gradients.".
	- It is unclear why "consistency" describes the magnitude of gradients.
9. At the end of Section 5.2 it reads: "However, Corollary 5.1 tells that we cannot both expand the class margin and shrink the class dispersion simultaneously. This is because $\Theta$ is moved in two exactly opposite directions for these two objectives. Thus, by maximizing the class margin and minimizing the class dispersion simultaneously, our loss guarantees that transformations along opposite directions cannot occur.".
	- This phrasing makes the two sentences read like a contradiction and should be rephrased for clarity.
10. The function class $H$ in Section 6.2 introduces a variety of of new parameters that introduce boundedness conditions. The paper should explain why these boundedness conditions are necessary.
11. The paragraph before Definition 6.3 references $L_{\text{clean}}$.  A reference to Equation 13 should be included for clarity.
12. Section 6.2 starts to denote inputs to the model as capital X, while before it was written in lower case. The notation should be held consistent throughout the manuscript.

**Audience:**

Yes

**Audience Explanation:**

The fundamental concepts underlying contrastive learning frameworks, i.e., learning class prototypes, maximizing intra-class compactness and inter-class margins, do not represent new ideas for training classifiers [4, 5, 6]. Likewise, the visual or theoretical analysis of how model robustness relates to hessian matrices has been addressed before [1, 2, 3]. However, this manuscript relates the three topics: (a) contrastive learning, (b) training with Gaussian input noise and (c) analysis of input loss curvature into one coherent framework. This unified perspective enables the authors to derive - to the best of the reviewer's knowledge - new theoretical insights, such as lower and upper bounds for loss curvature in terms of formerly defined hyperparameters for the contrastive learning framework.

Given the comprehensive nature of this work and its theoretical contributions, the work has the potential to spark interest in TMLR's audience.



[1]: Li, Hao, et al. "Visualizing the loss landscape of neural nets." _Advances in neural information processing systems_ 31 (2018).

[2]: Mustafa, Waleed, Robert A. Vandermeulen, and Marius Kloft. "Input hessian regularization of neural networks." _arXiv preprint arXiv:2009.06571_ (2020).

[3]: Dinh, Laurent, et al. "Sharp minima can generalize for deep nets." _International Conference on Machine Learning_. PMLR, 2017

[4]: Yang, Hong-Ming, et al. "Robust classification with convolutional prototype learning." _Proceedings of the IEEE conference on computer vision and pattern recognition_. 2018.

[5]: Wen, Yandong, et al. "A discriminative feature learning approach for deep face recognition." _European conference on computer vision_. Cham: Springer International Publishing, 2016.

[6]: Gupta, Rohit, et al. "Class prototypes based contrastive learning for classifying multi-label and fine-grained educational videos." _Proceedings of the IEEE/CVF conference on computer vision and pattern recognition_. 2023.

**Broader Impact Concerns:**

None.

**Claims And Evidence:**

Yes

**Claims Explanation:**

The authors pose three core claims:
- The defined loss function shall enforce **intra-class compactness**.
    - The authors derive theoretical evidence such as Proposition 5.1 or Theorem 6.2.  Additionally, t-SNE visualizations (Figures 11 and 12) illustrate compact clusters obtained using the proposed method.
- The loss function shall increase the **margin of embeddings to decision boundaries**.
    - To show this, the authors theoretically derive a lower bound for the margin between class centroids (Proposition 5.2). Experimentally, Section 7.2 shows that classifiers typically achieve higher test accuracies when trained with their loss function compared to other baseline methods, validating improved decision boundaries.
- The loss function shall enhance **model robustness**.
    - In Section 4.4 the authors discuss how the computation of class centroids can be stabilized across batches, that may have fluctuating noise ratios in practice, by introducing momentum terms.
    - Furthermore, Section 6.1 examines feature- and loss stability, relating these properties to upper and lower bounds for input loss curvature (Theorem 6.1), which in turn influences to classifier performance. Experimentally, the authors provide empirical evidence for this relation in Figures 7 and 8.

Overall, the author provide ample convincing theoretical and empiricial evidence in support of their core claims.

**Requested Changes:**

My requested changes mainly evolve around the mentioned weaknesses:
1. Please address the concern regarding the references that cannot be found.
2. Please improve the readability of the paper by clarifying the aspects noted in the additional comments.
3. Please either revise the proof or the claims made by Proposition 5.3.

---

> ### Author Response · Authors · 2025-11-11
> **Response by authors**
>
> First of all, we sincerely thank the reviewer for the thoughtful and detailed remarks, which help strengthen the paper. These aspects will be carefully addressed in the revised version. Below, we respond to each point.
>
> ## Erratums in references
> These errors occurred when merging different BibTeX drafts.
> - *Rong Jin et al.*, “Input Hessian Regularization for Adversarial Robustness”: this reference will be removed.
> - *Seyed-Mohsen Moosavi-Dezfooli et al.*, “A second-order approach to adversarial attacks and defenses”: will be corrected to " Robustness via curvature regularization, and vice versa. In Proceedings of the IEEE/CVF Conference on Computer Vision and Pattern Recognition"
>
> ## Convolution layer in Proposition 5.3
> The convolutional layer  here refers to the standard image convolutional layer. This will be clarified.
>
> ## Improving readability and clarifications
> 1. **Sections 5.2 and 6.2:** We will shorten both discussions. In Section 5.2, the Huygens Inertia Theorem reference will be removed as it is not really needed in this context. Section 6.2 will briefly summarize the main results, moving detailed analysis to the appendix.
> 2. **Equation 13:** We will clarify that two simultaneous forward passes are used for clean and noisy data (as shown in Fig. 2).
> 3. **Proposition 5.3:** The goal was to show that there exists theoretically a transformation that can modify either class margin or compactness without changing the model output. Although the practical dynamics are complex, our intent is to theoretically exclude such degenerate cases.
> 4. **Theorem 6.3:** We agree that it only provides a lower bound for the JSD between class distributions and does not imply increasing centroid separation. We will replace “separated” with “distinguishable,” which aligns with the meaning of JSD.
> 5. **Analogy to Huygens Theorem:** As noted in point (1), this analogy will be removed since it is not essential.
> 6. **Equations 8 and 9:** We will clarify that in $\delta_v$, v denotes *variance* and in $\delta_d$, d denotes *distance* (to decision boundaries).  This should help the readers to better follow the notation.
> 7. **Meaning of “closed cell”: by "closed cell", we refer to the **region in the feature space that is bounded by the decision boundaries separating one class from all others**. In the ideal case, these boundaries enclose a finite and well-defined region (a *closed cell*) corresponding to the domain of a specific class. However, in practice, depending on the parameter configuration of the softmax classifier, these boundaries may not form a fully closed region. We will clarify this in more detail in the paper.
> 8. **On “consistency”:** This refers to centroids being sufficiently updated, i.e., having gradients large enough to avoid becoming "outdated" during training. We will clarify this.
> 9. **Margin–dispersion clarification:** We thank the reviewer for pointing out this ambiguity. Our intended meaning is that expanding the class margin and shrinking the class dispersion according to the transformation $T_{\alpha}$ occur in opposite directions. Therefore, our combined loss formulation is designed to avoid this specific transformation (as this transformation not change the model’s performance). This is because moving in either of the two directions defined by $T_{\alpha}$ would either increase the class dispersion or decrease the class margin, which will not satisfy the joint optimization of both terms enforced by our constraints. This will be added to the paper.
> 10. **Boundedness parameters:** The boundedness parameters refer to the class centroids and the outputs of $f$. We explain this right after the statement:  "Such assumptions are natural in light of the regularization term in Eq. (11)." Specifically, the regularization term promotes that features lie within a reasonable range. Intuitively, if the feature space were unbounded, generalization would become extremely difficult, as features could fall anywhere in an infinite space. We will add this intuitive explanation to clarify our reasoning.
> 11. **Equation 13 reference:** Yes. We will add this.
> 12. **Notation consistency:**  Indeed, we use lowercase $x$ to denote a specific input sample (a realization), and uppercase $X$ to denote the random input variable following the input distribution. We will make this distinction clearer and ensure the notation is consistent throughout the paper.

---

> > ### Comment · Reviewer_hc69 · 2025-11-14
> >
> > I thank the authors for their thoughtful response. Correcting the references, rephrasing the claims of Proposition 5.3 and addressing my remarks about the readability of the paper in detail have improved my assessment of the manuscript.
> >
> > Regarding Theorem 6.3, my comment (No. 4) was not primarily about the choice of particular words such as "separated" or "distinguishable". Rather, it concerned the same question asked by Reviewer 2:
> > > Theorem 6.3: I am surprised that the result does not depend on the layer index $l$. Intuitively, I would expect the lower bound to increase with $l$. Is such a dependence hidden somewhere?
> >
> > Your reply to this has essentially clarified it:
> > > In fact, Theorem 6.3 provides a lower bound on the divergence across all layers, rather than the exact divergence for each specific layer. Hence, the bound represents the minimal value among all layers, which is why it does not explicitly depend on the layer index $l$.
> >
> > It may help readers if one or two sentences conveying this explanation are added directly below Theorem 6.3.

---

> > > ### Author Response · Authors · 2025-11-15
> > > **Response by authors**
> > >
> > > Yes, we will add this to the discussion below Theorem 6.3. Thank you again for your very thoughtful and detailed remarks.
> > >
> > > Kind regards,
> > >
> > > The authors

---

### Review · Reviewer_nSWX · 2025-11-03

**Summary Of Contributions:**

This work studies the robustness of a classification model.

First, the authors propose a loss function that operates in the feature space. This loss function encourages features belonging to the same class to be compact, while pushing features from different classes farther apart. The authors also propose injecting Gaussian noise into the data (similar to adversarial training) to further enhance the model’s robustness. This method consistently improves over baseline methods.

Second, the authors investigate the relationship between robustness under Gaussian noise and the curvature of the input loss. Roughly speaking, they show that robustness is equivalent to having a flat input loss landscape. The authors also conduct numerical experiments to verify this theoretical insight.

**Audience:**

Yes

**Audience Explanation:**

Yes. Robustness is an important aspect of neural network models, and this paper provides valuable techniques and insights.

**Broader Impact Concerns:**

There are no ethical concerns.

**Claims And Evidence:**

Yes

**Claims Explanation:**

I have read the main paper but not the appendix. Most of the claims made are well supported by clear evidence, either theoretically or empirically.

Below are a few claims that I found unconvincing:

- **Discussion in Section 5.2:** The authors theoretically justify that using one of their loss functions, either intra-class compactness or inter-class separability, is insufficient to ensure robustness. Hence, both loss functions are necessary. At the end of the section, it is stated that "*Thus, by maximizing the class margin and minimizing the class dispersion simultaneous, our loss guarantees that transformations along opposite directions cannot occur. That is, our loss can avoid a solution that does not improve the generalization error of the model.*"

    I think this claim is overstated. Corollary 5.1 only shows that both loss functions must be used at the same time, but it does not imply that using them together will always prevent solutions that fail to improve the model’s generalization error. I may have overlooked something, so please correct me if I am mistaken.

- **Discussion on Theorem 6.1 (Upper-bound and lower-bound of the input loss curvature):** The authors establish a connection between the input loss curvature and the model’s stability. It is stated that "*In Eq. (16), if we assume that the gradient of the loss at $x$ is sufficiently small, combining with the fact that $\sigma$ is generally small, we then have $\sigma^2\lVert \nabla l(x)\rVert^2\approx 0$. In this case,  $ 2\sum_i\lambda_i^2 + \left( \sum_i |\lambda_i| \right)^2 \gtrsim \frac{4(1-\eta)}{\sigma^4}(l_{\text{out}})-l(x))^2$.*"

   In fact, to obtain this last inequality, what we need is $(1-\eta)(l_{\text{out}}-l(x))^2 \gtrsim \sigma^2\lVert \nabla l(x)\rVert^2$. Should we expect this condition to hold? As far as I understand, in the idealized setting, the three terms $1-\eta$, $l_{\text{out}}-l(x)$ and $\lVert \nabla l(x)\rVert$ are all small, so it’s unclear to me whether the inequality $(1-\eta)(l_{\text{out}}-l(x))^2 \gtrsim \sigma^2\lVert \nabla l(x)\rVert^2$ is actually satisfied.

- **Theorem 6.3:** I am surprised that the result does not depend on the layer index $l$. Intuitively, I would expect the lower bound to increase with $l$. Is such a dependence hidden somewhere?

Minor comments on the experiments:
- In the experiments in Section 7.2, the highest noise level used for testing (in Table 1, Table 2, and Table 3) is roughly equal to the noise level used during training. I would be interested in seeing what happens when the test noise level exceeds the train noise level, as this could further demonstrate the model’s robustness under noise misspecification.

**Requested Changes:**

I would appreciate it if the authors could clarify the claims that I found unconvincing above.

---

> ### Author Response · Authors · 2025-11-09
>
> First of all, we thank you for your kindly thoughtful remarks. These remarks are interesting and will be carefully taken into consideration for the subsequent revised version. Here is our answer.
>
> > I think this claim is overstated. Corollary 5.1 only shows that both loss functions must be used at the same time, but it does not imply that using them together will always prevent solutions that fail to improve the model’s generalization error.
>
> In fact, in our phrasing *“our loss can avoid a solution that does not improve the generalization error of the model”*, the term "a solution" specifically refers to the transformation $T_{\alpha}$, not to all possible solutions that fail to improve performance.
>
> To avoid any misunderstanding, we will revise the sentence to read *“avoid this particular solution $T_{\alpha}$”* instead of "avoid a solution".
>
> We thank the reviewer for pointing this out and helping us improve the precision of our wording.
>
>
> > Discussion on Theorem 6.1 (Upper-bound and lower-bound of the input loss curvature): The authors establish a connection between the input loss curvature and the model’s stability... $(1 - \eta)\cdot \left(l_{\text{out}} - l(x)\right)^2 \gtrsim \sigma^2||\nabla l(x)||^2$.Should we expect this condition to hold?
>
> This remark is very insightful and interesting. In fact, in our theorem, we have two statements (1) and (2), which are made in two different senses.
>
> In statement (1), we show that in the *ideal setting* (i.e., when feature stability holds), the curvature is small.
>
> Statement (2) is in the opposite sense: here, we investigate the regime where feature stability or loss stability has **not yet** been achieved (i.e., when $(1 - \eta) \cdot (l_{\text{out}} - l(x))^2$ is sufficiently large). We aim to understand what happens if we can somehow reduce the input loss curvature in this regime. Specifically, we show that reducing the input loss curvature implies either feature or loss stability.
>
> We will add this clarification to make it clear to readers that statement (2) concerns the regime where stability is not yet established, i.e., when $(1 - \eta) \cdot (l_{\text{out}} - l(x))^2$ remains large. Thank you for this helpful remark.
>
> > Theorem 6.3: I am surprised that the result does not depend on the layer index. Intuitively, I would expect the lower bound to increase with $l$. Is such a dependence hidden somewhere?
>
> In fact, Theorem 6.3 provides a *lower bound on the divergence across all layers*, rather than the exact divergence for each specific layer. Hence, the bound represents the minimal value among all layers, which is why it does not explicitly depend on the layer index $l$.
>
> Our goal was to derive a **model-agnostic** lower bound that holds universally, without relying on architectural details. However, we agree that, under additional assumptions about the layer type or network structure, one could potentially obtain **sharper, layer-dependent bounds**. This is indeed an interesting direction for future work, though it lies beyond the scope of the current paper.
>
>
> > Minor comments on the experiments: In the experiments in Section 7.2, the highest noise level used for testing (in Table 1, Table 2, and Table 3) is roughly equal to the noise level used during training. I would be interested in seeing what happens when the test noise level exceeds the train noise level.
>
> This is a very interesting remark, as it allows us to examine how the model learns to "extrapolate" stability beyond the noise level seen during training.
>
> To address this, we added experiments with higher noise levels: **20/255** for CIFAR-10, **42/255** for SVHN, and **60/255** for the road data — noting that the original highest training noise levels were **16/255**, **36/255**, and **40/255**, respectively.
>
> It is important to mention that at these levels, the noise already alters the semantic content of the inputs quite drastically, so testing at even higher noise levels would become unreasonable.
>
> The results are presented in the table below. We observe that the model maintains fairly good performance even when evaluated at noise levels beyond those used in training, which is a promising indication of its robustness. These results will be integrated into the corresponding experimental tables for completeness.
>
>
> |                       | Normal training | Only on noisy data | On both clean and noisy data | Stability training | Ours |
> |-----------------------|-----------------|--------------------|------------------------------|--------------------|------|
> | CIFAR10 (20/255)      | $50.32 \pm 0.29$ | $87.918 \pm 0.16$ | $89.04 \pm 0.21$ | $84.62 \pm 0.21$ | $\textbf{89.42} \pm 0.11$ |
> | SVHN (42/255)         | $58.87 \pm 0.16$ | $90.39 \pm 0.11$ | $90.43 \pm 0.08$ | $89.71 \pm 0.07$ | $\textbf{90.86} \pm 0.08$ |
> | Road Data (60/255)    | $43.81 \pm 1.02$ | $92.05 \pm 0.84$ | $92.92 \pm 0.46$ | $90.87 \pm 0.69$ | $\textbf{93.65} \pm 0.49$ |

---

> > ### Comment · Reviewer_nSWX · 2025-11-30
> >
> > I appreciate the authors' detailed response. All of my concerns have been resolved.
> >
> > * **Discussion in Section 5.2:** Yes, please revise the manuscript accordingly.
> > * **Theorem 6.1 (Upper-bound and lower-bound of the input loss curvature):** Thank you for the detailed reply. I now understand the theorem and its implications. Please add an explanation of the second statement (Eq. (16)) to improve clarity. I think this is a good result.
> > * **Theorem 6.3:** I see. The response "Hence, the bound represents the minimal value among all layers, which is why it does not explicitly depend on the layer index." clarifies my question. I suggest adding this remark after Theorem 6.3 to avoid any confusion.
> > * **Additional experiments:** The result is good. I suggest including them into the manuscript.

---

### Review · Reviewer_KYC7 · 2025-11-03

**Summary Of Contributions:**

The authors propose to append several complementary "loss" terms to the training objective of deep neural network classifiers in order to improve their predictive robustness (resilience to noise). Interestingly, all these terms rely on the centroid of each class representation as given by the penultimate layer of the neural network (denoted $m_c$ for $c = 1,...,C$).
- The "intra-class compactness" term, borrowed from De Brabandere et al. (2017), enforce every representation of the sample sharing a same class $c$ to be close to $m_c$
- An "inter-class separability" term enforcing a large margin between class centroids,
- An optional "regularization"  term penalizing the norms centroid vectors $m_1,...m_C$.
- A "noisy data feature alignment" term ensuring that the representations of perturbed inputs of samples of class $c$ are close to the centroid $m_c$ (computed solely on clean data).

The manuscript clearly explains how the authors have achieved combining these terms to keep the accuracy of neural network classifiers on clean data while making them robust to noisy samples. On the downside, the complementariness of each term could be better explained, by clarifying some passages and providing ablation studies.

**Audience:**

Yes

**Audience Explanation:**

Despite not being very knowledgeable of the relevant literature, the proposed method seems new. I believe it can be added to the list of tools one can implement to improve the robustness of their neural network.

**Broader Impact Concerns:**

-

**Claims And Evidence:**

Yes

**Claims Explanation:**

The rationale of each design choice is well explained. The notation is clean. In general, the paper provides both intuitive explanations and theoretical insights, as well as a convincing empirical assessment of the method. However, there are some aspects that could be improved.

**1. Discussion on margin-based methods**

Enforcing the margin between classes is an historic principle in machine learning, and few are said about this in the paper. The SVM is only mentioned when citing Tang (2013) in the related work section, even if the expression "hinge loss" appears before Definition 5.1. I would like to see a short discussion about the relation of the proposed method with the SVM objective.

Also, it is quickly mentioned after Eq. (9) that the proposed  "inter-class separability" margin loss is inspired by the work of Elsayed et al. (2018). It would be important to highlight the key differences. Note that Elsayed et al. (2018) discussed the relation of their approach with the  SVM margin. The current manuscript only claims that "[maximizing the] distance from each sample to all the decision boundaries" would be "very costly and not really necessary" (Section 4.1), which is elusive and deserves explanations.

**2. On the complementarity of intra-class compactness and inter-class separability**

I think Section 5.2 is an important justification for the proposed method, but is difficult to read. The first sentence ("Recall that the Huygens Inertia Theorem decomposes total inertia into"...) deserves a reference and further explanations. The implications of Proposition 5.3 and Corrollary 5.1 could be explained more gently. I suggest explaining $\mathcal{T}_\alpha$ before the proposition (is it a *linear* map? Also, it might be better to rename $\alpha$ to avoid confusion with Eq. (7)'s hyperparameter). A figure could possibly help the reader here!

**3. Empirical role and complementariness of all loss terms**

The manuscript lacks training details, particularly about the values of hyperparameters $\alpha$, $\beta$ and $\lambda$ multiplying the training objective loss components. Ideally, I would like to see illustrations of the classification accuracy when varying these hyperparameters, which would also lead to ablation studies for the cases $\alpha=0$, $\beta=0$ and $\lambda=0$.

**Requested Changes:**

I consider the points numbered **1** to **3** critical.

Also, here are minor comments:
- Section 4: Please state from the beginning that the proposed "loss" terms are intended to be added to a typical empirical loss term in the overall training objective
- For the reader's benefit, I suggest recalling close to Eq. (9) that $\mathcal{P}_{ci}$ is defined by Eq. (6)
- Section 7: There are some incorrect usages of the citation commands. e.g.:
    - CIFAR-10 (Krizhevsky et al. (2009))  --> CIFAR-10 (Krizhevsky et al., 2009)
    - ResNet18 He et al. (2016) --> ResNet18 (He et al., 2016)
- Section 7.1; paragraph "Neural network models': It is unclear if the "backbone networks" (ResNet18 and MobileNetV3) are frozen or fine-tuned during the experiments. The sentence segment "followed by some fully connected layers" could be replaced with an exact statement.

Last but not least:
- The references section needs to be proofread. Some published peer-reviewed works are cited as arXiv preprints (e.g., Foret et al. is an ICLR 2021). Conferences are sometimes referred to by their full names and sometimes by their abbreviations (e.g., NeurIPS and Advance in neural information processing systems).

---

> ### Author Response · Authors · 2025-11-08
> **Response by authors**
>
> First of all, the authors sincerely appreciate your kindly thoughful reviews. These aspects will be added to the paper.
>
> ## 1.  Discussion on margin-based methods
> ### Relation to SVMs.
> Classical SVMs, and the deep SVM variant of Tang, maximize the margin between classes through a hinge-loss objective defined on separate linear classifiers. Specifically, Tang replaces the softmax layer by K one-vs-rest SVMs, each introducing additional trainable weights and slack variables. This setup enforces the margin at the classifier output level but does not directly constrain the geometry of the features. In contrast, our approach retains the standard softmax layer and instead defines, analytically, the decision boundaries induced by this softmax classifier in the feature space. This enables us to impose margin-based constraints and intra-class compactness directly on the learned features without introducing additional classifiers or slack parameters.
> ### Relation to Elsayed et al. (2018).
> This work only focuses on the approximated euclidean margin to decision boundaries, and have no constraints on the intra-class compactness. For this, they maximize the distance of each individual example to the decision boundaries. In contrast, our method only maximize the distance between the class centroid and the decision boundaries, reducing signifcantly the computations. This is because we have already the class compactness forcing each points to be not far from class centroid. Note also that the intra-class compactness is important for promoting smaller curvature, as explained in theoretical part.
>
> This will be carefully added to the discussion and related works.
>
> ### 2. On the complementarity of intra-class compactness and inter-class separability.
> ### About Huygens Inertia Theorem.
> We also have another reviewer recommending to remove this if not really needed. While this is quite standard in classical ML, this is not really necessary in modern Deep Learning. So we shall remove this part, and instead focus on the explanation about the transformation, as you recommend, for better clarity for readers.
> > I suggest explaining $\mathcal{T}_{\alpha}$ ...rename $\alpha$.
>
> Thank you for this kind recommend. We do provide the detail about this transformation in the full proof, but we shall add brief discussion about this on the main text. Basically it has a linear impact on a subset of the model parameters and the other part does not change.
> ### rename $\alpha$: yes, we shall replace by $\nu$.
>
> ### 3.  Empirical role and complementariness of all loss terms.
> > The manuscript lacks training details.
>
> For simplicity, we fix $\alpha=\beta=1$ and $\gamma_{reg}=10^{-3}$ for all the experiments. This detail will be added to the paper.
>
> ### About ablation study.
> For this, we have added an ablation study. In fact, the constraint on noisy data is similar to the class compactness constraint on clean data. Hence, to really see the "ablation role" of inter-class separability and intra-class compactness constraints, we use only clean data constraints, to avoid a form of "cheating" for the intra-class compactness.
>
> First of all, we fix $\gamma_{reg}=10^{-3}$ and set 3 scenarios ($\alpha=\beta=1$), ($\alpha=0$, $\beta=1$),  ($\alpha=1$, $\beta=0$). Then, we also fix $\alpha=\beta=1$, and use 2 scenarios $\gamma_{reg}=0$ and $\gamma_{reg}=1$ to see the role of regularization. We have the following table (recall that $\alpha$ for compactness and $\beta$ for separability). Some important remarks:
> - Using both compactness and separability constraints is better than using only each of them.
> - Compactess constraint alone is better than separability constraint alone. This is aligned with our discussion where we point out that softmax loss helps to produce separable features, but no guarantee about class compactness. So compactness constraint is likely to be more imporant.
> - Fixing $\alpha=\beta=1$, using regularization is likely to perform better than not using it (i.e. $\gamma_{reg}=0$). Practical advice: if we do not want to tune, using small value (like 0.001) is a good choice, it performs well accross the 2 ablations.
>
> This will be carefully added to the paper.
>
> | Hyper-params | α=1, β=0, γ_reg=0.001 | α=0, β=1, γ_reg=0.001 | α=1, β=1, γ_reg=0.001 | α=1, β=1, γ_reg=0 | α=1, β=1, γ_reg=1 |
> |-------------|-----------------------|-----------------------|-----------------------|-----------------|-----------------|
> | CIFAR10     | 93.43                 | 92.25                 | 94.01                 | 93.66           | 93.81           |
> | SVHN        | 96.15                 | 95.98                 | 96.64                 | 96.55           | 96.67           |
>
>
> ### About minor comments:
> Thank you for your careful comments about citations, reference to equations,  loss details and model structure. We do not have space to write the details here, but these are rather simple to address, that we will carefully re-check.

---

> > ### Comment · Reviewer_KYC7 · 2025-11-17
> >
> > I acknowledge that the authors seriously considered my comments. I think the discussion will benefit from the added discussions. Additionally, the ablation study will help readers better understand the benefits of each component of the proposed framework.

---

### Decision · Action_Editor_ZSUV · 2025-12-10

**Recommendation:** Accept as is

**Audience:**

Yes

**Audience Explanation:**

This paper studies the robustness of deep neural networks, which can attract lots of interest from many researchers.

**Claims And Evidence:**

Yes

**Claims Explanation:**

Yes, the claims made in the submission are strongly supported by accurate, convincing, and clear evidence. The authors provide a rigorous theoretical framework with formal propositions and theorems, accompanied by complete mathematical proofs in the appendices. These theoretical insights are systematically validated through extensive experiments on standard benchmarks and a custom dataset, using multiple baselines and a variety of input perturbations. The results are presented quantitatively with detailed tables and qualitatively through visualizations like t-SNE plots, which consistently demonstrate the method's effectiveness in improving robustness while maintaining clean-data accuracy.